# Kinetics of HIV-1 capsid uncoating revealed by single-molecule analysis

Chantal L Márquez[1,2], Derrick Lau[1,2], James Walsh[1,2], Vaibhav Shah[1,2], Conall McGuinness[1,2], Andrew Wong[3], Anupriya Aggarwal[3], Michael W Parker[4,5], David A Jacques[1], Stuart Turville[3], Till Böcking[1,2]*

[1]EMBL Australia Node in Single Molecule Science, School of Medical Sciences, UNSW, Sydney, Australia; [2]ARC Centre of Excellence in Advanced Molecular Imaging, UNSW, Sydney, Australia; [3]The Kirby Institute, Sydney, Australia; [4]Australian Cancer Research Foundation Rational Drug Discovery Centre, St. Vincent's Institute of Medical Research, Melbourne, Australia; [5]Department of Biochemistry and Molecular Biology, Bio21 Molecular Science and Biotechnology Institute, University of Melbourne, Melbourne, Australia

**Abstract** Uncoating of the metastable HIV-1 capsid is a tightly regulated disassembly process required for release of the viral cDNA prior to nuclear import. To understand the intrinsic capsid disassembly pathway and how it can be modulated, we have developed a single-particle fluorescence microscopy method to follow the real-time uncoating kinetics of authentic HIV capsids in vitro immediately after permeabilizing the viral membrane. Opening of the first defect in the lattice is the rate-limiting step of uncoating, which is followed by rapid, catastrophic collapse. The capsid-binding inhibitor PF74 accelerates capsid opening but stabilizes the remaining lattice. In contrast, binding of a polyanion to a conserved arginine cluster in the lattice strongly delays initiation of uncoating but does not prevent subsequent lattice disassembly. Our observations suggest that different stages of uncoating can be controlled independently with the interplay between different capsid-binding regulators likely to determine the overall uncoating kinetics.
DOI: https://doi.org/10.7554/eLife.34772.001

*For correspondence:
till.boecking@unsw.edu.au

**Competing interests:** The authors declare that no competing interests exist.

## Introduction

HIV is an enveloped virus that carries its RNA genome and associated viral proteins within a protein shell called the capsid (*Welker et al., 2000*). Upon engagement of CD4 and the chemokine receptor CCR5 or CXCR4, the viral membrane fuses with the plasma membrane of the host, depositing the viral core (defined here as the capsid and its contents) in the cytoplasm (*Blumenthal et al., 2012*). In order to establish infection, the virus must reverse transcribe its single stranded RNA genome into double stranded DNA, traverse the cytoplasm and cross the nuclear membrane, after which it integrates into the host chromosome (*Bukrinsky, 2004*). We now know that the capsid plays a key role in these processes and is critical for successful infection. It not only acts as a shield to protect the viral genomic material from pattern recognition and degradation (*Lahaye et al., 2013*; *Rasaiyaah et al., 2013*), but is also thought to facilitate reverse transcription (*Jacques et al., 2016*), engage with the nuclear pore complex (*Burdick et al., 2017*; *Dharan et al., 2016*; *Matreyek et al., 2013*), and direct integration site targeting (*Ocwieja et al., 2011*; *Schaller et al., 2011*; *Sowd et al., 2016*). In order to achieve many of these proposed functions, the capsid must interact with host proteins and small metabolites, as well as disassemble to release the viral DNA at the appropriate place and time (*Campbell and Hope, 2015*). The nature of this 'uncoating' process and how it is influenced by host co-factors remains a key unanswered question in HIV biology.

**eLife digest** Viruses need to enter their host's cells in order to replicate their genetic material and produce more copies of the virus. A protein shell called a capsid protects the virus during this journey. But the structure of the capsid presents a mystery. How can this protein shell be strong enough to remain intact as it enters a host cell, and yet quickly open up to release the viral genome after replication?

Unlike the capsids of many other viruses, those of HIV have irregular structures that rapidly fall apart once removed from the virus. This has thwarted attempts to study intact HIV capsids in order to understand how they work. However, we do know that HIV hijacks a range of molecules produced by the invaded host cell. Dissecting their effects on the capsid is key to understanding how the capsid disassembles.

Márquez et al. have now developed a method that can visualize individual HIV capsids – and how they disassemble – in real time using single-molecule microscopy. This revealed that capsids differ widely in their stability. The shell remains closed for a variable period of time and then collapses catastrophically as soon as it loses its first subunit.

Using the new technique, Márquez et al. also found that a small molecule drug called PF74 causes the capsid to crack open rapidly, but the remaining shell is then stabilized against further disassembly. These observations reconcile seemingly contradictory observations made by different research groups about how this drug affects the stability of the capsid.

The method developed by Márquez et al. enables researchers to measure how molecules produced by host cells interact with the viral capsid, a structure that is fundamental for the virus to establish an infection. It could also be used to test the effects of antiviral drugs that have been designed to target the capsid. The new technique has already been instrumental in related research by Mallery et al., which identifies a molecule found in host cells that stabilizes the capsid of HIV.
DOI: https://doi.org/10.7554/eLife.34772.002

The HIV capsid comprises 1000–1500 copies of a single viral protein, CA, corresponding to less than half of the total CA released by proteolytic processing of the Gag polyprotein during maturation (*Briggs et al., 2004*). CA is a two-domain protein, which can form both hexameric and pentameric assemblies. The conical capsid is a fullerene structure, which incorporates 12 pentamers into the otherwise hexameric lattice, a geometric constraint that must be satisfied in order to achieve capsid closure (*Li et al., 2000*; *Pornillos et al., 2011*). However, there is no such constraint on the number of hexamers nor the positions of pentamers and, as such, HIV capsids are polymorphic. Typically, capsids have a conical shape with an average length of 119 nm and an average width of 61 nm (*Briggs et al., 2003*). In addition, a large fraction of viral particles contain 'defective' capsids, which do not have closed surfaces (*Frank et al., 2015*; *Mattei et al., 2016*; *Yu et al., 2013*). This high degree of polymorphism and poor fidelity of assembly mean that the HIV capsid is a technically challenging entity to study, especially when compared to the more regular icosahedral viruses, many of which can even be crystallized.

Current biochemical methods used to study HIV capsid stability include an in vitro uncoating assay for the release of CA from isolated capsids (*Ambrose and Aiken, 2014*; *Shah and Aiken, 2011*), the fate of capsid assay (*Stremlau et al., 2006*; *Yang et al., 2014*) and the cyclosporin A washout assay (*Hulme et al., 2011*). These assays measure uncoating in bulk, that is rely on observing the average behavior of large numbers of viral cores. Fluorescence microscopy methods that track the uncoating process in fixed or live cells (*Francis et al., 2016*; *Mamede et al., 2017*) can resolve steps in the uncoating process of individual capsids in the cytoplasm but complementary in vitro methods that allow high throughput measurements and detailed kinetic studies under defined conditions are still missing.

Here we describe an in vitro single-molecule fluorescence imaging assay that allows us to follow the uncoating kinetics of hundreds of individual HIV capsids in a single experiment. This type of single-molecule analysis has the advantage that it can resolve intermediates in the disassembly pathway that are otherwise averaged out in traditional ensemble assays. By observing the properties of many individual capsids, it is possible to classify them according to their uncoating behaviors. This versatile

method enables 'bottom-up' approaches to determine the effect of individual host molecules and drugs and is also compatible with 'top-down' studies in which the influence of whole-cell lysates can be observed. Using this method, we are able to classify virions into three categories based on their uncoating behavior. We have also been able to define two discrete uncoating events that we term 'capsid opening' and 'lattice disassembly' and show that known cofactors and drugs have different effects on these two processes. This observation has allowed us to resolve the ambiguity as to what is meant by 'uncoating' and how it can be influenced by external factors.

## Results

### Single-particle fluorescence imaging of capsid opening

Fusion of the viral particle with the plasma membrane marks the point in time when the viral core is first exposed to the cytoplasm, but the effects of cellular proteins and small molecules on the stability of the capsid lattice are largely unknown. Here, we designed a fluorescence imaging assay to pinpoint the time of capsid opening at the single-particle level after exposing capsids to biochemically different environments in vitro. We produced viral particles containing GFP as a solution phase marker using a proviral construct with Gag-internal GFP (*Figure 1A*) (*Aggarwal et al., 2012*; *Hübner et al., 2007*). GFP is expressed as part of the Gag polyprotein and released by proteolysis during maturation, whereby a fraction of GFP molecules are compartmentalized within the viral capsid, with the remainder enclosed outside the viral capsid but within the viral membrane (*Mamede et al., 2017*; *Yu et al., 2013*). As a control we also used constructs containing a mutation in the late domain of Gag that leads to a block in the abscission of viral particles from the producer cell. The number of GFP-positive particles in the cell supernatant decreased by 99.7% when viral release was blocked (*Figure 1B*), confirming that essentially all GFP-positive particles released in the absence of the block represent viral particles. The fluorescent viral particles were biotinylated, purified by gel filtration and captured via streptavidin onto the surface of a coverslip modified with an inert polymer layer that prevents non-specific adsorption of viral particles and proteins (*Figure 1B*). Using microfluidics, we then delivered a solution containing the bacterial pore-forming protein perfringolysin O (PFO). PFO efficiently permeabilized the viral membrane by assembling into characteristic ring-shaped membrane pores with a mean pore diameter of about 35 nm (*Figure 1D and E*), consistent with its activity on cholesterol-containing membranes (*Dang et al., 2005*). These pores are sufficiently large to permit the passage of proteins while the viral core is retained within the perforated viral membrane allowing the core to be observed over time while it can undergo disassembly.

In our assay, we detected viral membrane permeabilization and capsid opening using time-lapse total internal reflection fluorescence (TIRF) microscopy to monitor the release of GFP molecules trapped in these compartments. GFP-loaded viral particles captured onto the coverslip appeared as bright diffraction-limited spots in the fluorescence image (*Figure 1F*), whereby a typical field of view contained several hundred to a thousand particles with a broad distribution of GFP intensities (*Figure 1G*). We extracted GFP intensity traces for each particle by summing the total fluorescence intensity above background of the diffraction limited spot in each frame of the TIRF movie (see *Figure 1H* for selected traces and for snapshots of the corresponding particles over time). A common feature of the vast majority of traces was a large drop in the GFP signal that occurred from one frame to the next at a variable time point after addition of PFO to the flow channel. We attributed this sudden drop in signal to the release of the pool of GFP molecules enclosed by the viral membrane as a result of PFO-mediated permeabilization. This process was not synchronized and typically occurred at random times for individual particles over a period of several minutes, as expected for the stochastic process underlying assembly of PFO into membrane pores. The precise time of permeabilization could, nevertheless, be determined from the large drop in the GFP fluorescence intensity traces (indicated by 'P' in *Figure 1H*) obtained by single-particle analysis.

Tracking of the residual GFP signal after permeabilization then permitted us to distinguish between three different classes of particles with 'leaky', 'opening' or 'closed' capsids. The majority of traces was characterized by a complete loss of GFP fluorescence upon membrane permeabilization (*Figure 1H*, left panel, see also *Figure 2A*). These particles were classified as containing leaky capsids with a defective and/or unstable CA lattice that did not retain the pool of GFP inside the

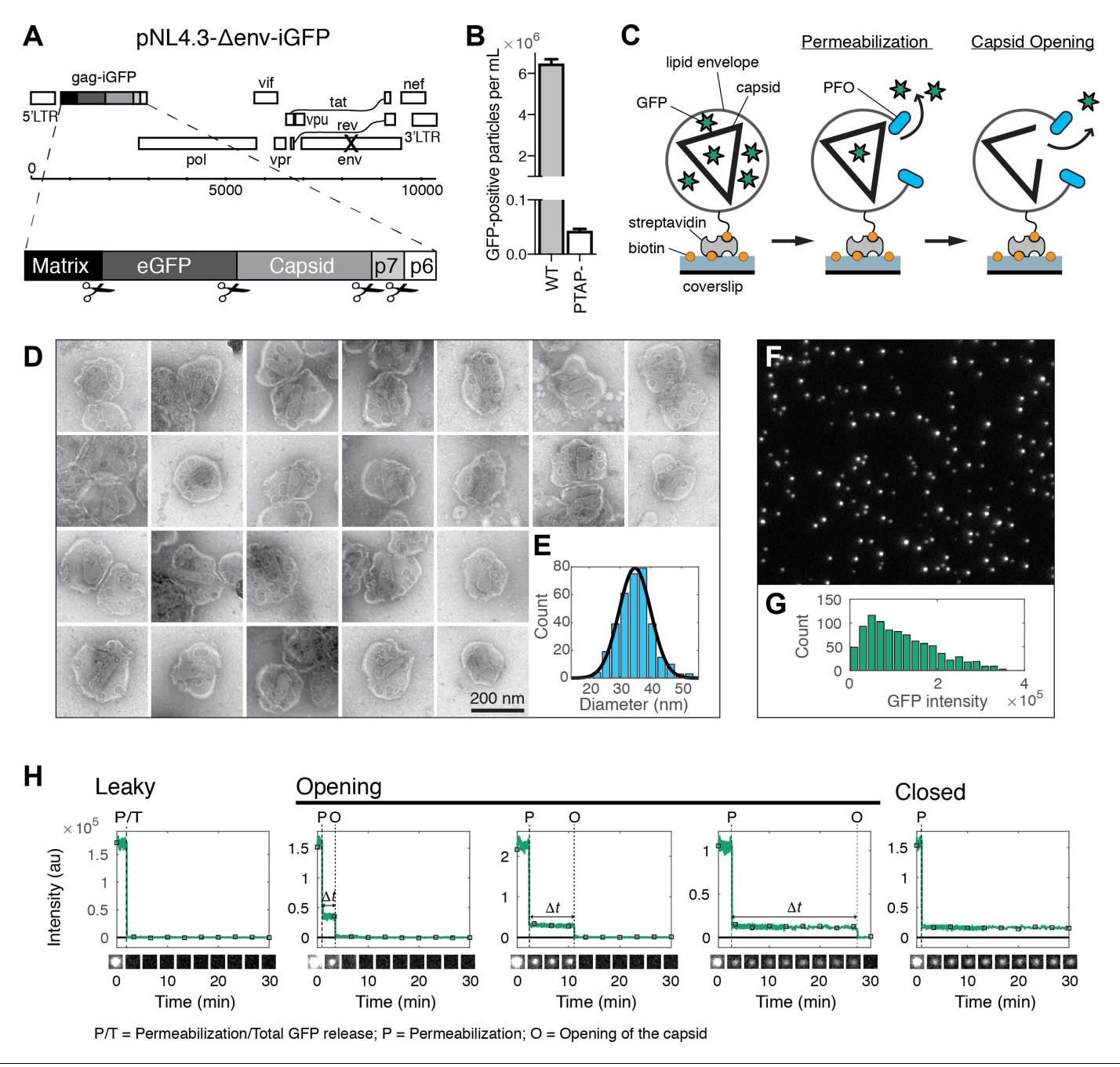

**Figure 1.** HIV capsid opening assay. (**A**) Map of the proviral DNA contained in the vector pNL4.3-iGFP-ΔEnv. Viral particles with Gag-internal GFP (released as a solution phase marker by proteolysis during maturation) were generated by transfection of HEK293T cells with pNL4.3-iGFP-ΔEnv and psPAX2 in a molar ratio of 1.4:1. (**B**) GFP-positive particles released from HEK293T cells transfected with either wild type or PTAP motif mutants of pNL4.3-iGFP-ΔEnv and psPAX2. (**C**) Schematic diagram of the TIRF assay to measure capsid opening. (**D**) Gallery of negative staining TEM images of viral particles incubated with the pore-forming protein PFO. (**E**) Distribution of PFO pore diameters (34.8 ± 5.2 nm, mean ±standard deviation, $N = 347$). (**F**) TIRF image of immobilized viral particles in a 450 × 370 pixel region of the field of view. (**G**) Distribution of GFP intensity for each particle in the field of view. (**H**) Fluorescence intensity traces of viral particles with leaky, opening and closed capsids showing the release of GFP contained by the envelope at the point of envelope permeabilization ('P') with PFO and release of encapsidated GFP after spontaneous opening ('O') of the HIV capsid. The capsid opening time (Δt) is the time from permeabilization to capsid opening. The black squares indicate the frames at which the snapshots shown below each trace were taken. The TIRF movie was recorded with an imaging frequency of 1 frame s$^{-1}$.

DOI: https://doi.org/10.7554/eLife.34772.003

The following figure supplement is available for figure 1:

*Figure 1 continued*

**Figure supplement 1.** Histogram of the fraction of encapsidated GFP.
DOI: https://doi.org/10.7554/eLife.34772.004

capsid (*Welker et al., 2000*; *Yu et al., 2013*). The remainder of traces showed a residual GFP signal after membrane permeabilization that constituted on average 13% of the total GFP signal prior to permeabilization (*Figure 1—figure supplement 1*). This fraction was consistent with the amount of GFP expected to be randomly trapped inside the volume of the capsid (relative to the volume of the viral particle) during maturation. Loss of this residual GFP signal to background levels at a later point in time was attributed to the spontaneous opening of the capsid lattice to create a hole sufficiently large to permit the passage of GFP (indicated by 'O' in *Figure 1H*, middle panels). We then calculated the capsid opening time (denoted as Δt in *Figure 1H*) for each trace as the difference between the time of membrane permeabilization and the time of capsid opening. In a fraction of traces the residual GFP signal remained constant until the end of the TIRF movie and the corresponding particles were classified as having closed capsids that retain GFP for the duration of the experiment (*Figure 1H*, right panel). We conclude that monitoring the steps of GFP release from viral particles at the single-particle level reveals differences in capsid opening times.

## Effect of capsid-binding molecules on capsid opening kinetics

We used the capsid opening assay to characterize the intrinsic stability of the HIV capsid in the absence and presence of molecules known to affect the stability of the CA lattice. *Figure 2A* shows the fraction of capsids that are leaky, undergo opening or remain closed within 30 min of adding PFO to the immobilized particles. Leaky capsids constituted the largest fraction of particles (60–80%), irrespective of which type of capsid-binding molecules were present in the solution. Even in the absence of capsid-binding molecules, we observed variability in the fraction of leaky capsids in this range for different batches of particles. These observations suggest that leaky particles result from assembly defects during maturation, which are unaffected by modulators of CA lattice stability that the capsid encounters at a later stage (e.g. after entry into the cell).

To obtain an estimate of the half-life of the opening process, we analyzed the distribution of capsid opening times determined for hundreds of particles that undergo opening (*Figure 2B, C*). The survival curve of intact capsids (i.e. those that remain positive for encapsidated GFP) exposed to buffer without additives could be approximated with an exponential decay function (*Figure 2B*, black lines and *Figure 2—source data 1*) yielding an estimate for the capsid half-life of 8 min (*Figure 2C*), which is similar to a previous estimate of 10 min for the half-life of capsid uncoating in vitro (*Francis et al., 2016*). Fitting a bi-exponential decay made it possible to quantify a sub-population of capsids (~20%) that were substantially less stable and opened with a half-life of less than 1 min (*Figure 2B*, dashed grey line and *Figure 2—source data 2*).

To validate that the GFP release kinetics depend on capsid stability, we assessed the effect of the E45A mutation in CA on capsid opening. This mutation is known to yield hyperstable capsids (*Forshey et al., 2002*; *Yang et al., 2012*), which was also apparent in our capsid opening assay with a 4.5-fold increase in half-life compared to capsids assembled from wild-type CA (*Figure 2C*).

The ability to identify different subpopulations of capsids and to extract a half-life for capsid opening allowed us to quantitatively compare the effect of known capsid-binding proteins and small molecules on capsid stability. Cyclophilin A (CypA) is a host cell protein that binds to the cyclophilin binding loop of CA exposed on the exterior of the capsid (*Gamble et al., 1996*), and this interaction is thought to control capsid stability after cell entry (*Shah et al., 2013*). In our assay, CypA had no effect on capsid half-life when added at a concentration of 1 μM and lead to an approximately two-fold increase in half-life at 20 μM (*Figure 2C*).

The small molecules PF74 and BI2 bind to hexameric CA and inhibit HIV infection at an early post-fusion event (*Bhattacharya et al., 2014*; *Blair et al., 2010*; *Lamorte et al., 2013*; *Price et al., 2014*). Addition of these inhibitors to the capsid opening assay at high concentrations within the range typically used in cellular and biochemical assays revealed an increase in the ratio of opening to closed particles after 30 min (*Figure 2A*) and a pronounced acceleration of capsid opening with half-lives of 41 s and 2.4 min, respectively (*Figure 2C*). In contrast, the small molecule

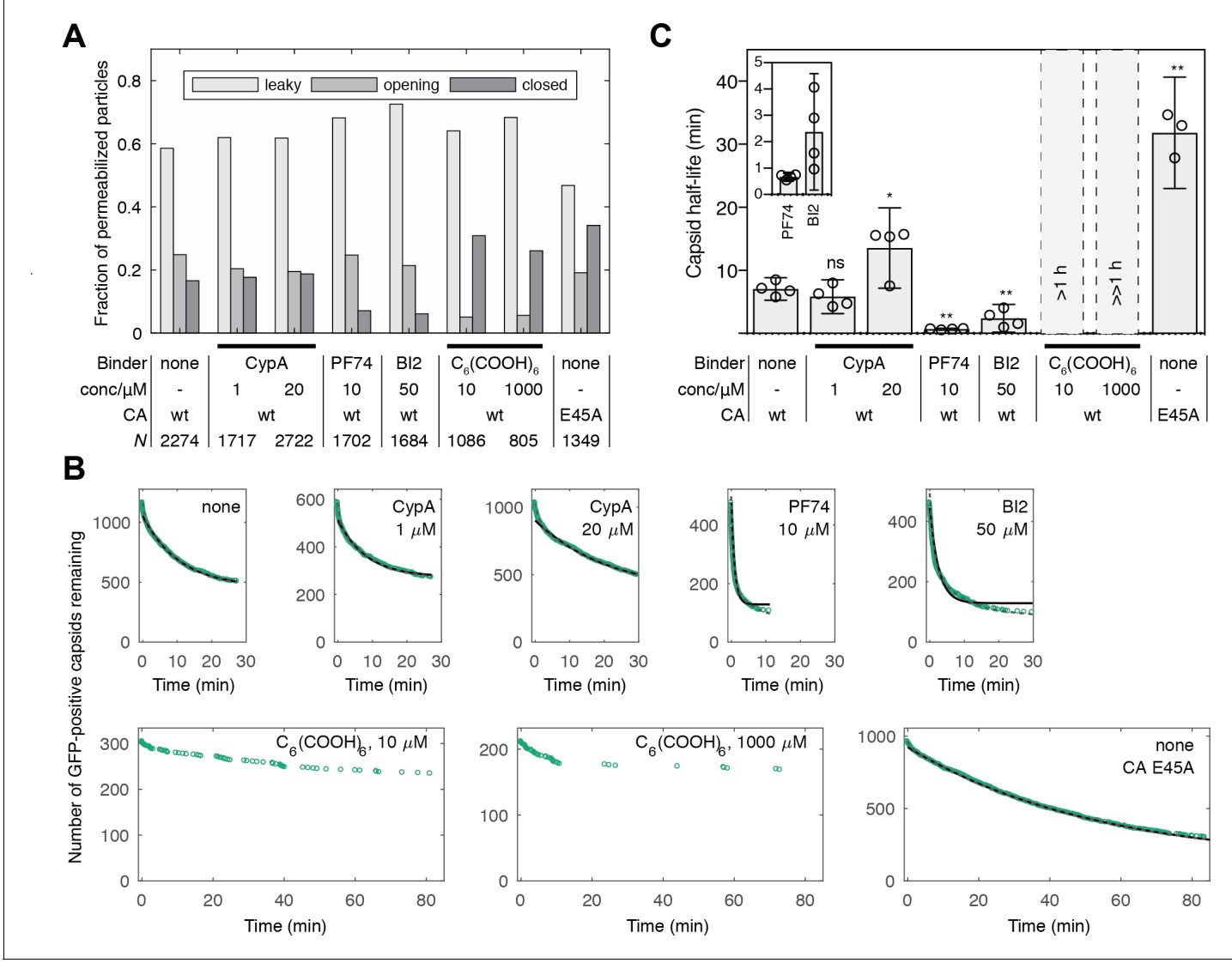

**Figure 2.** Kinetics of capsid opening. (A) Bar chart showing the fraction of capsids classified as 'leaky', 'opening' and 'closed' that are present 30 min after addition of PFO to permeabilize the envelope, recorded in the absence and presence of capsid binders. (B) Survival curves of capsids with encapsidated GFP after permeabilization with mono- (black line) and bi-exponential (dotted grey line) fit. The parameter values of the fits are provided in *Figure 2—source data 1* and *2*. The data in panels A and B was combined from multiple measurements (total number of repeats/number of viral preparations): No binder (3/2); CypA - 1 µM (4/2); CypA - 20 µM (5/2); PF74 - 10 µM (4/2); BI2 - 50 µM (5/3); hexacarboxybenzene - 10 µM (3/2); hexacarboxybenzene - 1000 µM (2/1); CA E45A (5/3). (C) Half-lives of intact capsids determined by fitting survival curves from individual repeats with a mono-exponential decay function. The error bars represent 95% confidence intervals; unpaired t-test with Welch's correction, p=0.2797 (1 µM CypA), p=0.0403 (20 µM CypA), p=0.0403 (20 µM CypA), p=0.0014 (10 µM PF74), p=0.0022 (50 µM BI2), p=0.0043 (CA E45A). The half-lives of capsids in the presence of hexacarboxybenze ($C_6(COOH)_6$) could not be determined with certainty and were estimated to be >1 hr.

DOI: https://doi.org/10.7554/eLife.34772.005

The following source data is available for figure 2:

**Source data 1.** Table of coefficient values from a mono-exponential fit of the survival curves in *Figure 2B*.
DOI: https://doi.org/10.7554/eLife.34772.006

**Source data 2.** Table of coefficient values from a bi-exponential fit of the survival curves in *Figure 2B*.
DOI: https://doi.org/10.7554/eLife.34772.007

hexacarboxybenzene ($C_6(COOH)_6$), which binds to hexameric CA (*Jacques et al., 2016*), showed the opposite effect by strongly inhibiting capsid opening (*Figure 2A*). There was an insufficient number of opening events to allow a robust fit of the survival curves but we estimated that the half-lives in the presence of 10 and 1000 µM hexacarboxybenzene were in the order of 1 hr and 10 hr, respectively. Taken together, our observations show that measuring GFP release from authentic viral capsids at the single-particle level is suitable to reveal the heterogeneity of capsid assembly states and quantify the effect of capsid binders on the opening kinetics of intact capsids.

## Kinetics of protein binding to intact HIV capsids

The half-life of the HIV capsid is sufficiently long to use PFO-permeabilized viral particles as a platform to quantify the binding of labeled proteins to authentic viral capsids at the single-particle level as shown in *Figure 3A*. To demonstrate this approach, we added CypA labeled with Alexa Fluor 568 (AF568) at the same time as PFO to the flow channel and followed the fluorescence signals of GFP packaged into the virion and AF568-CypA using dual-color TIRF microscopy. For binding analysis, we selected particles with capsids that remained intact (GFP-positive) for a sufficient period of time to obtain a binding equilibrium. Using a calibration value obtained by single-molecule photobleaching, we converted the AF568 intensity to obtain the number of CypA molecules bound at each location corresponding to a closed capsid.

Representative traces of CypA binding to closed capsids are shown in *Figure 3B*. As before, PFO-mediated permeabilization of each individual particle was evident from the large drop in GFP intensity to a level corresponding to the pool of GFP trapped inside the closed capsid. Coincident with GFP signal loss, we observed a rapid increase in the CypA signal up to an equilibrium binding level. We then obtained the CypA binding kinetics for the entire population of closed capsids in the field by aligning all single-particle traces at the time of membrane permeabilization. The aligned traces recorded in the presence of 1 µM CypA are shown as a heatmap in *Figure 3C*. The median binding trace showed that binding equilibrium was rapidly established, suggesting that the PFO-permeabilized membrane does not present a significant barrier for diffusion of proteins to and from the capsid. The number of molecules bound at equilibrium in the experiment shown in *Figure 3C* had a median of 73 molecules per capsid. Across 12 independent experiments, we measured $58 \pm 8$ molecules per capsid in the presence of 1 µM CypA (*Figure 3—figure supplement 1*). When imaged for longer periods of time, the plateau level increased slowly, suggesting that a low level of irreversible or non-specific binding may occur over time. However, binding of CypA was completely abolished in the presence of the competitive inhibitor cyclosporin A, which binds to the binding pocket on CypA (*Figure 3C*, bottom panels) (*Mikol et al., 1993*). This observation indicates that the interaction with the permeabilized viral particle is largely driven by an interaction with sites recognized by its binding pocket.

Next, we used the binding assay to obtain an estimate of the dissociation constant ($K_D$) for the CypA-capsid interaction and the number of binding sites that can be occupied simultaneously on the assembled capsid. Mean binding curves were obtained from aligned single-particle traces recorded at concentrations ranging from 0.1 to 20 µM (*Figure 3D*). Analysis of the equilibrium binding levels gave an estimate of the $K_D$ of 10–12 µM with an estimated 500–1000 molecules bound per capsid at saturation (*Figure 3E* and *Figure 3—figure supplement 2*). This number of sites represents roughly half of the number of CA proteins in an assembled capsid (*Briggs et al., 2003*). The expected CypA:CA ratio at saturation is also 1:2, because binding to both cyclophilin loops of CA molecules connecting neighboring hexamers in the lattice would lead to steric clashes (*Liu et al., 2016*). This observation suggests that the lattice is completely coated with CypA at saturation, and that binding sites on different locations of the capsid are equivalent.

Finally, we obtained the rate of CypA dissociation by measuring the decay of the mean fluorescence intensity associated with closed capsids after removing CypA from solution by flushing the flow channel with buffer (*Figure 3F*). The majority of the fluorescence signal (85%) disappeared rapidly with a rate of 1.5 $s^{-1}$. This rate was close to the rate of solution exchange achievable in our microfluidic set-up, such that the off-rate of CypA measured here may be limited by mass transport. The residual CypA signal associated with the particles, possibly representing irreversibly bound CypA, decayed with a rate consistent with photobleaching. For comparison, we also measured the binding and dissociation of unlabeled CypA to CA and cross-linked CA hexamers using surface plasmon resonance. Equilibrium binding analysis of the SPR data gave a $K_D$ of 21.8 µM and 19 µM for

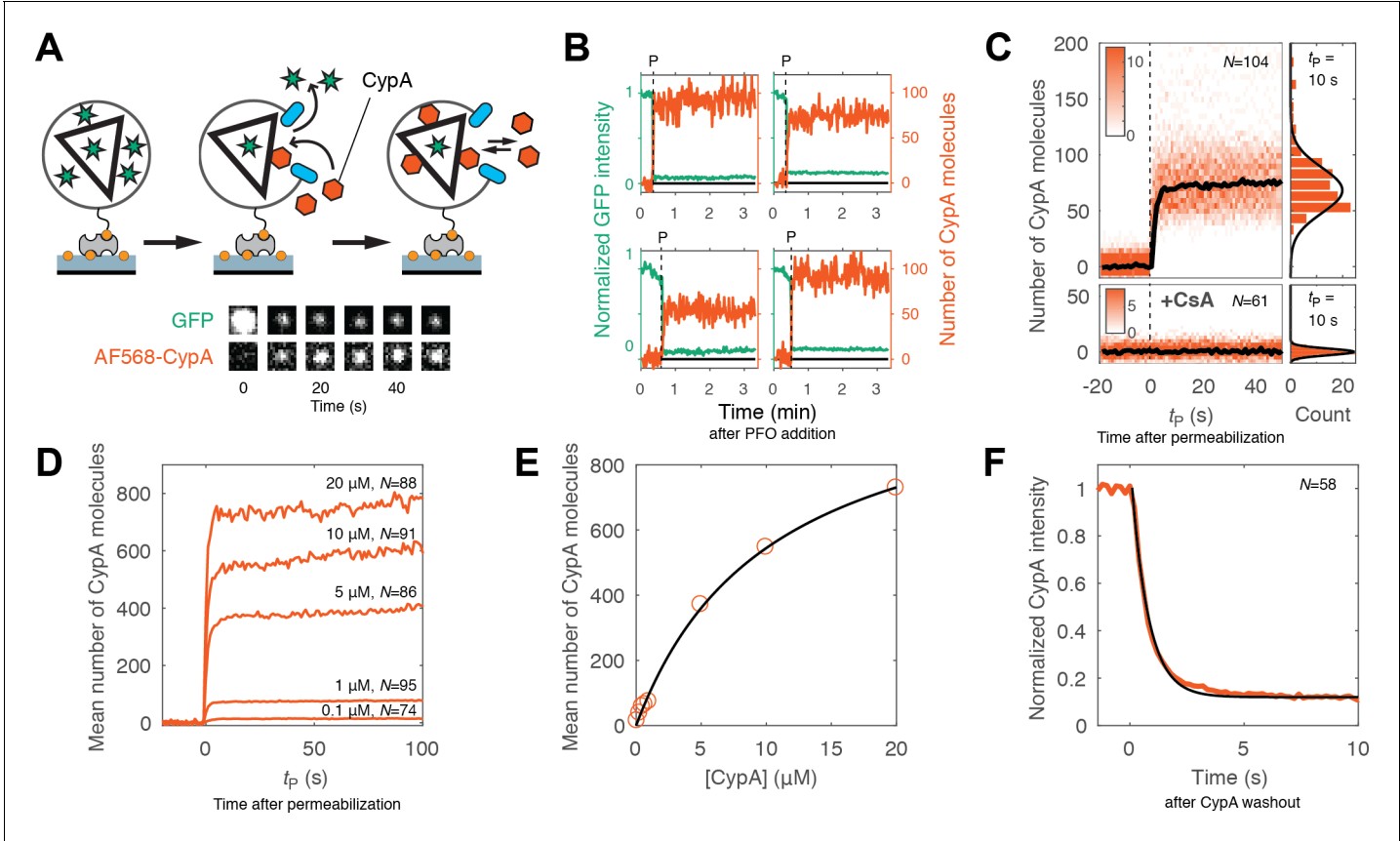

**Figure 3.** Kinetics of CypA binding to the intact capsid. (A) Schematic diagram of the TIRF assay for measuring protein binding to the intact capsid and snapshots of the GFP signal and AF568-CypA signal of a selected viral particle at different times. (B) Example traces of CypA binding (vermillion) to individual closed capsids that retain GFP (green) after permeabilization of the viral envelope (note the residual GFP signal after the large drop). The concentration of CypA was 1 µM. (C) CypA binding traces (vermillion) to all closed capsids in the field of view aligned with respect to the time of envelope permeabilization ($t_P$ = 0). The bold black line represents the median binding trace. CypA binding was recorded in the absence (top) and presence (bottom) of cyclosporine A (CsA). The histograms at the right show the distribution of intensities after binding equilibrium is reached (at $t_P$ = 10 s). Gaussian fitting gave a mean ±standard deviation of 73 ± 20 and 1.6 ± 2.9 in the presence and absence of CsA, respectively. Representative data from four (−CsA) and two (+CsA) independent experiments using different viral preparations. (D) Median CypA binding traces recorded at CypA concentrations ranging from 0.1 to 20 µM as indicated above the corresponding trace. Binding experiments with concentrations above 1 uM were carried out with a mixture of unlabeled and labeled CypA, whereby the concentration of the labeled CypA was kept at 1 µM. (E) Median number of CypA molecules bound at equilibrium as a function of CypA concentration (vermillion circles) and fit of an equilibrium binding model (black line). The fit gave the following estimates for the interaction: KD = 10 µM; number of CypA-binding sites on the intact capsid ~1000. See *Figure 3—figure supplement 2* for an independent repeat experiment. (F) Mean CypA intensity measured at closed capsids after wash-out of CypA from the flow channel at $t$ = 0. The black line represents a fit of the data with an exponential decay function.

DOI: https://doi.org/10.7554/eLife.34772.008

The following figure supplements are available for figure 3:

**Figure supplement 1.** Variation in the median number of molecules bound at equilibrium per closed capsid measured in independent experiments at different CypA concentrations.
DOI: https://doi.org/10.7554/eLife.34772.009
**Figure supplement 2.** Equilibrium binding analysis of CypA binding to capsid.
DOI: https://doi.org/10.7554/eLife.34772.010
**Figure supplement 3.** SPR measurements of the CypA-CA interaction.
DOI: https://doi.org/10.7554/eLife.34772.011

interaction with immobilized CA (consistent with previous data [*Yoo et al., 1997*]) and CA hexamer (*Figure 3—figure supplement 3*), respectively, while the off-rates determined after removal of CypA from solution were 4 ± 0.3 s$^{-1}$ and 3.5 ± 0.4 s$^{-1}$, respectively. We conclude that CypA binds with low affinity to the authentic viral capsid lattice shortly after membrane permeabilization in an

interaction that is governed by rapid kinetics and not fundamentally different from its interaction with CA that is not part of a lattice.

## Visualizing capsid lattice disassembly using CypA paint

The capsid opening assay pinpoints the start of the uncoating reaction. To also measure the kinetics of lattice disassembly beyond the appearance of the first defect, we explored the possibility of using labeled CypA to 'paint' the CA lattice, whereby the signal of bound CypA is proportional to the number of CA molecules that remain at a given point in time during disassembly, similar to a previous approach using CypA-DsRed (*Francis et al., 2016*). Our observations above revealed that CypA fulfils two important requirements for its use as a capsid lattice paint. Firstly, CypA binding does not have a measurable effect on capsid opening when used at concentrations of up to 1 µM (*Figure 2C*). Secondly, CypA binds (and dissociates) with fast rates to binding sites all over the surface of the lattice (*Figure 3*). In the experiments shown here, we added labeled CypA to the buffer at the same time as adding PFO, but in principle it is possible to add (and withdraw) the label at any point in time during the uncoating experiment. We then observed the CypA signal as a function of time for each GFP-positive particle in the field of view, whereby the vast majority of particles showed the appearance of a CypA signal above background shortly after permeabilization. Below we focus on the analysis of leaky and opening capsids, whereby the intensity of the CypA signal was converted into a measure for the integrity of the capsid lattice.

Particles with leaky capsids (complete disappearance of the GFP signal in a single step) showed a spike in the CypA signal upon membrane permeabilization that typically decayed rapidly (within seconds to minutes, see *Figure 4B* for example traces). The median CypA intensity trace obtained by aligning all traces with leaky capsids at the time of membrane permeabilization showed instantaneous accumulation of CypA intensity followed by a short dwell time of 5–7 s and exponential decay characterized by a half-life of 14–25 s (*Figure 4C*). We interpreted this signal as follows: particles with leaky capsids contain a CA lattice that is poised to fall apart and only persists for a short period of time before disassembly. The distribution of CypA molecules bound per capsid at the intensity peak (*Figure 4C*, middle panel) suggested the presence of particles containing almost complete capsids with only minor defects as well as particles with only partially assembled lattices. The distribution of CypA paint intensities measured 5 min after permeabilization (*Figure 4C*, right panel) revealed a small residual signal, suggesting the presence of residual CA.

Single-particle traces of opening capsids (*Figure 4D*) were characterized by the following phases: First, rapid binding of CypA upon membrane permeabilization, reaching a binding equilibrium while the capsid remained closed (*Figure 4E*, left heatmap). Second, the CypA fluorescence signal started to decay within a short dwell time of capsid opening (GFP release) and then reached a stable level above background. We interpret the second phase as the disassembly of most of the CA lattice. The median disassembly kinetics after capsid opening (*Figure 4E*, right heatmap) were governed by a half-life of approximately 70–100 s. The rapid decay of the lattice after initiation of uncoating observed here with labeled monomeric CypA is consistent with previous measurements in vitro and in cells using a tetrameric probe, CypA-dsRed, that binds irreversibly to the capsid lattice (*Francis et al., 2016*). As observed for leaky capsids, we detected residual CypA intensity at the end of the disassembly phase, possibly due to the presence of some CA molecules remaining associated with the viral RNA inside the permeabilized viral particle.

Overall, the disassembly process after capsid opening followed a similar process as observed for leaky capsids, but with slower kinetics. We conclude that capsid opening is the rate-limiting step for capsid uncoating, and lattice disassembly will proceed rapidly once the capsid has developed the first defect.

## CypA paint reveals the effects of PF74 and hexacarboxybenzene on capsid lattice disassembly

As shown above, PF74 accelerated capsid opening, while hexacarboxybenzene strongly inhibited this step. A simple explanation of these observations is that binding of these compounds globally affects capsid stability. In this case, we would predict that the rates of lattice disassembly are affected in the same way as capsid opening, that is increased in the presence of PF74 and decreased in the presence of hexacarboxybenzene. To test this prediction, we used CypA paint to determine

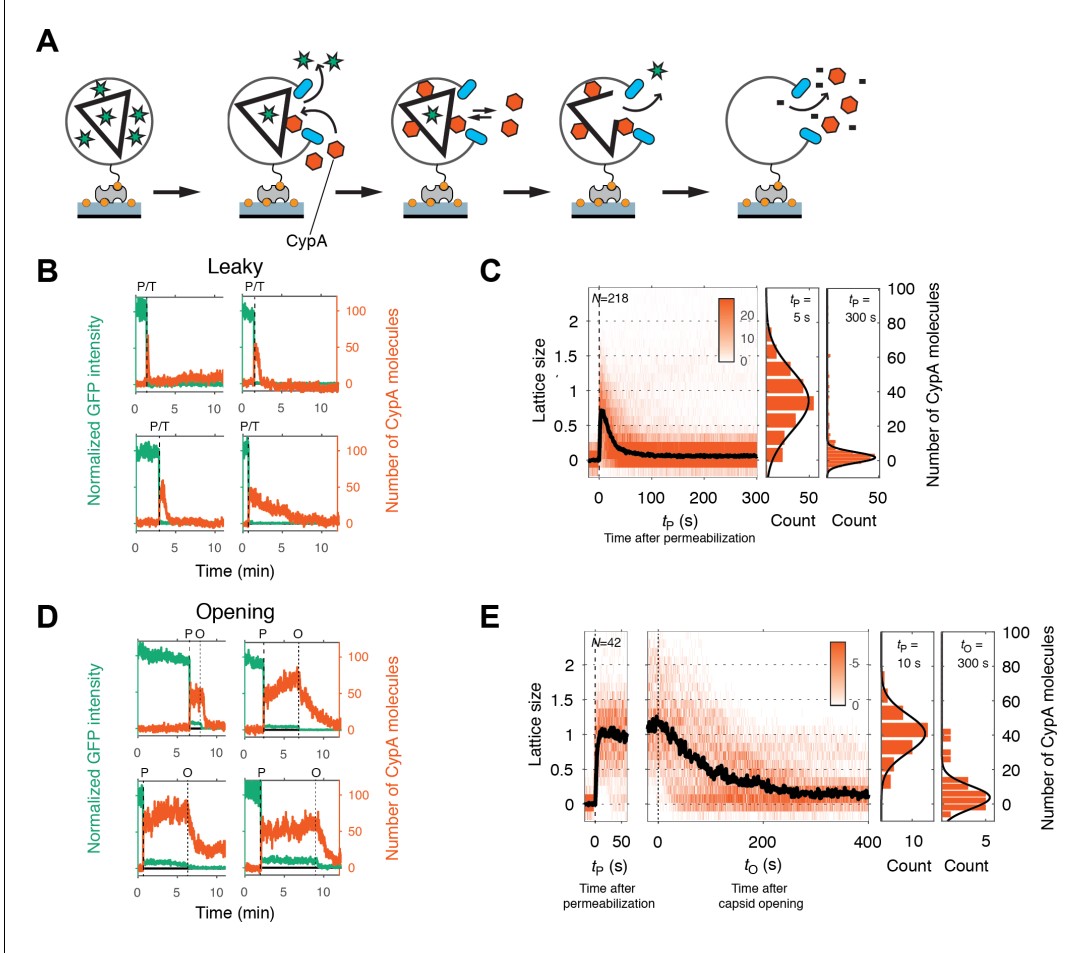

**Figure 4.** Painting the capsid with CypA to visualize lattice disassembly kinetics. (**A**) Schematic diagram of the TIRF CypA paint assay. (**B**) Example traces of AF568-CypA binding (vermillion) to individual leaky capsids that do not retain GFP (green) after permeabilization of the viral envelope. The concentration of CypA was 0.7 μM. (**C**) Heatmap of CypA binding traces to all leaky capsids in the field of view aligned with respect to the time of envelope permeabilization ($t_P = 0$). The bold black line represents the median CypA trace. The histograms at the right show the distribution of intensities at $t_P = 5$ s and 300 s. (**D**) Example traces of AF568-CypA binding (vermillion) to individual capsids that initially retain a fraction of the GFP signal (green) after permeabilization of the viral envelope, and then open to release this residual GFP signal. The concentration of CypA was 0.7 μM. (**E**) Heatmap of CypA intensity traces of all capsids in the field of view that undergo opening. The traces in the left panel are aligned with respect to the time of envelope permeabilization ($t_P = 0$), while the traces in the right panel are aligned with respect to capsid opening. The bold black line represents the median CypA trace. The histograms at the right show the distribution of intensities after binding equilibrium is reached (at $t_P = 10$ s) and at the end of lattice disassembly. Representative data from three independent experiments using different viral preparations.

DOI: https://doi.org/10.7554/eLife.34772.012

the effect of these compounds on the kinetics of lattice disassembly. CypA binding to the capsid was indistinguishable in the presence and absence of these compounds (*Figure 3—figure supplement 1*), suggesting that binding of either compound does not interfere with CypA binding. Surprisingly, in the presence of PF74, we observed no decrease in the CypA paint signal after capsid opening (*Figure 5B*, right heatmap). This observation suggests that the lattice develops a defect to allow the passage of proteins but is otherwise highly stabilized such that no (further) CA proteins are released from the lattice. The amount of CA protein lost during the opening step was below the limit of detection in our assay (compare the CypA paint intensity histograms before and after capsid opening, *Figure 5B*). This stabilization of the lattice was also observed for leaky capsids (*Figure 5A*). Here, too, the CypA paint signal remained constant for at least 8 min. The observations indicate that the lattice did not release a substantial number of individual CA subunits (or small multimers), but we cannot rule out breakage of the capsid into stable parts that are too large to traverse the PFO pores. In contrast to the strong stabilization observed with PF74, hexacarboxybenzene only had a

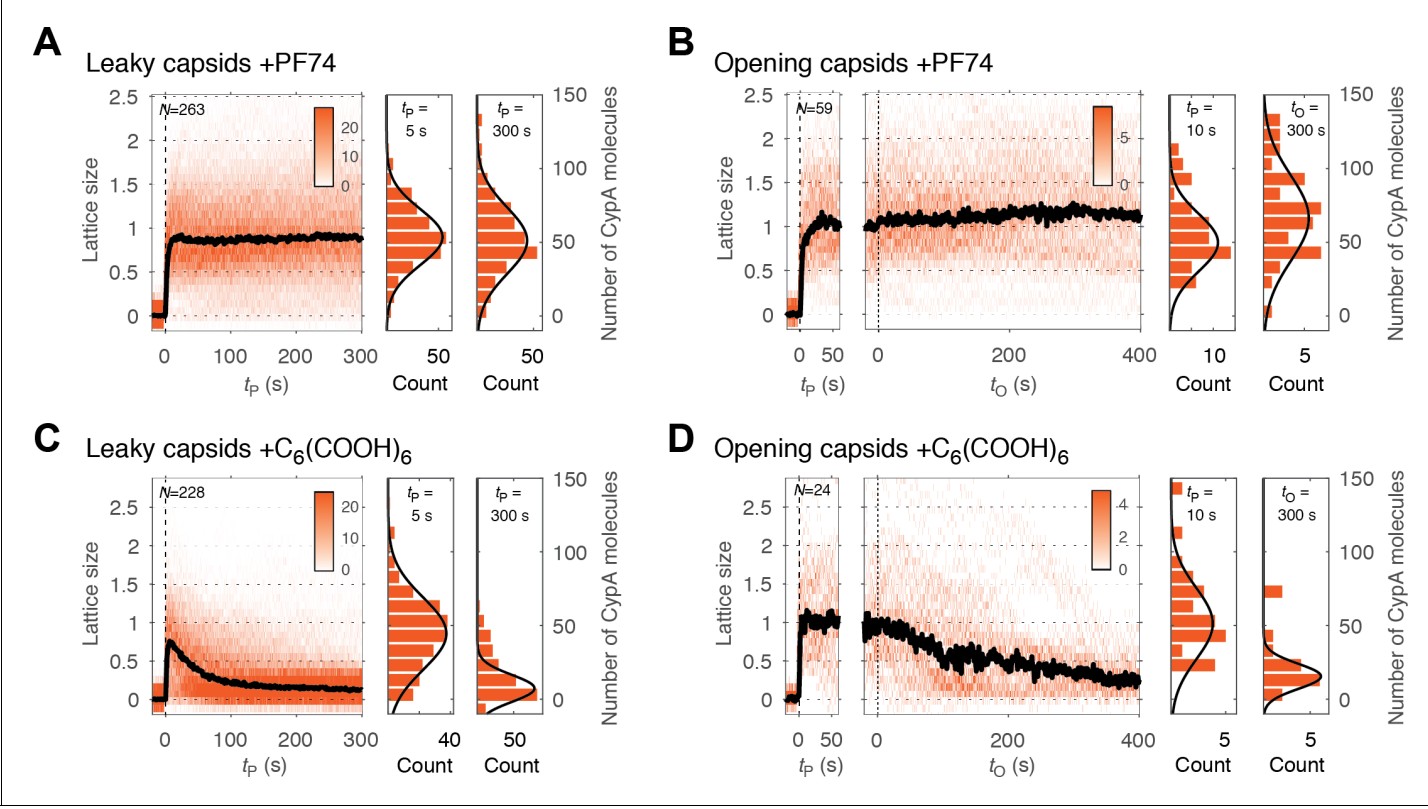

**Figure 5.** PF74 but not hexacarboxybenzene stabilizes the CA lattice after capsid opening. (A, C) Heatmaps of CypA binding traces to all leaky capsids in the field of view aligned with respect to the time of envelope permeabilization ($t_P$ = 0) in the presence of PF74 (A) or hexacarboxybenzene (C). The bold black line represents the median CypA trace. (B, D) Heatmaps of CypA intensity traces of all capsids in the field of view that undergo opening in the presence of PF74 (B) or hexacarboxybenzene (D). The traces in the left panel are aligned with respect to the time of envelope permeabilization ($t_P$ = 0), while the traces in the right panel are aligned with respect to capsid opening ($t_O$ = 0). The bold black line represents the median CypA trace. Heatmaps are representative of data acquired in two independent experiments using different viral preparations.

DOI: https://doi.org/10.7554/eLife.34772.013

modest stabilizing effect on the CA lattice after capsid opening, as seen from the CypA paint signal decay from that point onwards (*Figure 5D*); the inability to prevent collapse of the lattice was also apparent for leaky capsids (*Figure 5C*). We conclude that while PF74 promotes a structural change leading to a defect in the capsid that allows passage of proteins, only little (if any) CA protein is lost during this step, and the remaining lattice is highly stabilized. In contrast, hexacarboxybenzene strongly inhibits capsid opening but has little effect on uncoating once the process has started.

## Cell lysate slows capsid opening but not lattice disassembly

Finally, we conducted capsid opening and CypA paint experiments in the presence of cell lysate to test whether it was compatible with our single-particle approaches. PFO-mediated permeabilization was unaffected when HeLa cell lysate was added to the buffer at a final protein concentration of 1 mg mL$^{-1}$. However, capsid opening was slowed down with a four-fold increase in the half-life of this process (*Figure 6A and B*). CypA bound to the exposed capsid in the same manner as observed in buffer (*Figure 6C*). The number of CypA molecules bound on intact (closed) capsids at equilibrium was comparable to that measured in buffer (*Figure 3—figure supplement 1*), indicating that there was no competition with endogenous CypA due to the high dilution factor used in this experiment. CypA paint revealed that the process of uncoating in diluted cell lysate was in principle the same as is buffer, with the lattice falling apart shortly after it had opened up, albeit with a slightly reduced rate (*Figure 6D and E*). The stabilizing effect exerted by the diluted cell lysate is consistent with previous observations (*Francis et al., 2016*; *Fricke et al., 2013*; *Guth and Sodroski, 2014*), and it may be amplified when components are present at cytosolic concentrations. Identification of components

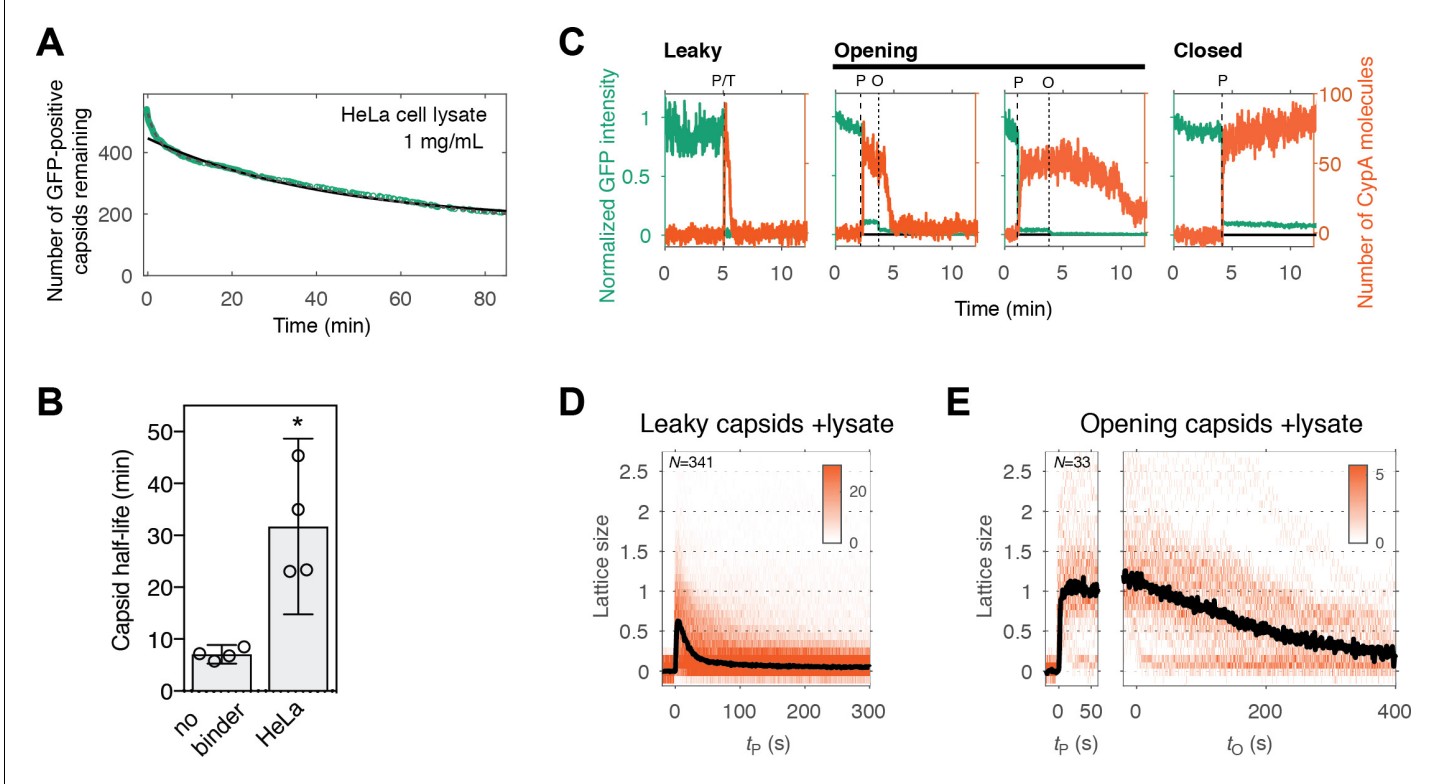

**Figure 6.** Capsid opening and lattice disassembly measured in the presence of HeLa cell lysate. (A) Survival curve of GFP-positive capsids in the presence of HeLa cell lysate. The half-life determined from the curve is 48 min. Combined data from a total of five repeats using two different viral preparations. (B) Half-life of intact capsids determined by fitting survival curves from individual repeats with a mono-exponential decay function. The error bars represent 95% confidence intervals; unpaired t-test with Welch's correction, p=0.0184. (C) Example traces of AF568-CypA binding (vermillion) in the presence of HeLa cell lysate to individual capsids (green) after permeabilization of the viral envelope. The concentration of CypA was 1 µM. (D) Heatmap of CypA binding traces to all leaky capsids in the field of view aligned with respect to the time of envelope permeabilization. The bold black line represents the median CypA trace. (E) Heatmap of CypA intensity traces of all capsids in the field of view that undergo opening. The traces in the left panel are aligned with respect to the time of envelope permeabilization, while the traces in the right panel are with respect to capsid opening. The bold black line represents the median CypA trace.

DOI: https://doi.org/10.7554/eLife.34772.014

The following source data is available for figure 6:

**Source data 1.** Table of coefficient values from a mono-exponential fit of the survival curves in *Figure 6A*.
DOI: https://doi.org/10.7554/eLife.34772.015
**Source data 2.** Table of coefficient values from a bi-exponential fit of the survival curves in *Figure 6A*.
DOI: https://doi.org/10.7554/eLife.34772.016

contributing to this effect is the subject of future investigation, whereby the single-particle methods introduced here are suitable to dissect the effect of specific perturbations in the context of a complex mixture such as cell lysate.

## Discussion

Measurement of the single-particle uncoating kinetics of authentic HIV capsids immediately after removing the protective barrier of the viral membrane has allowed us to observe the effects of capsid-binding molecules on discrete steps in the capsid uncoating process. The order of events established with our new assays would have been obscured in ensemble measurements due to dephasing. We observed the following main characteristics of HIV-1 capsid uncoating and its interactions with capsid-binding molecules: (1) The integrity and stability of capsids in a population is highly heterogeneous, whereby the majority of capsids contain defects and/or uncoat immediately after removal of the viral membrane. (2) In the absence of capsid binders, uncoating is an all-or-none

process that proceeds rapidly after the first defect is created. (3) Capsid-binding molecules exhibit differential effects on the steps that control initiation and continuation of uncoating.

The structural heterogeneity of the HIV capsid is well described, with the assembly process yielding a variety of conical, tubular and other shapes that often contain structural defects such as gaps in the CA lattice (*Briggs et al., 2003*; *Frank et al., 2015*; *Mattei et al., 2016*; *Yu et al., 2013*). Here, we identify functionally distinct classes of capsids on the basis of their GFP release kinetics, allowing us to quantify the functional heterogeneity for hundreds of capsids within a population and relate them to different assembly states. By utilizing a pore-former to permeabilize the viral membrane in a controlled manner while recording capsid opening traces, we are able to observe early events that occur upon exposure of the capsid to a change in conditions and thus capture unstable assembly states. This approach circumvents the need for isolating viral cores, a process which exposes cores to experimental buffer conditions for extended periods of time (*Shah and Aiken, 2011*) and would result in a loss of this information. Our results show that the assembly states range from incomplete or imperfect (leaky) lattices to fully assembled (closed) capsids that nevertheless exhibit pronounced differences in uncoating kinetics. Here, we have quantified the fraction of particles of different stability for a viral particle that lacks HIV envelope protein and has GFP inserted into the Gag polyprotein. These features may affect maturation and capsid assembly or lead to misclassification of some particles, for example as a result of incomplete proteolysis. It is therefore important to note that wild type HIV may exhibit a different distribution of capsid stability types. Nevertheless, it is striking that 60–80% of capsids in our experiments are not fully closed and are unable to retain GFP upon membrane permeabilization. This category includes early assembly intermediates but mostly structures that are essentially fully assembled (as judged by our CypA paint method, see below) yet unable to form a closed cone. We assign the latter class of leaky structures to capsids with holes that are also observed in intact viral particles by cryo-electron microscopy (*Mattei et al., 2016*) and by GFP release in vivo (*Yu et al., 2013*). The abundance of these types of particles obtained here using a viral preparation protocol that avoids harsh treatments (such as centrifugation) suggests that capsid assembly frequently becomes trapped at this stage, whereby remodeling of the lattice to allow it to close might be a slow process.

Host cell proteins that interact with the viral capsid after cell entry often recognize assembled CA rather than CA monomers (*Yamashita and Engelman, 2017*). Cross-linked CA hexamers and self-assembled lattices from recombinant CA (*Ganser-Pornillos et al., 2004*; *Pornillos et al., 2010*) or CA-NC protein (*Campbell and Vogt, 1995*) are powerful tools for identification and structural studies of capsid-binding molecules but do not fully recapitulate the structural features (including curvature and flexibility) of the lattice that may be required for binding. In contrast, our assay provides access to the authentic, wild-type capsid lattice for determining interaction kinetics, affinity and stoichiometry, and thus assess structural models for the binding mode of host proteins on the intact lattice. A recent study of self-assembled CA lattices in complex with CypA using cryo-electron microscopy and solid-state NMR spectroscopy suggested that each CypA bridges two hexamers in the lattice by simultaneously contacting the cyclophilin loops on two neighboring CA proteins, one via its canonical binding site and one via a postulated non-canonical binding site located on the opposite side of the CypA molecule (*Liu et al., 2016*). While our binding measurements are consistent with a 1:2 CypA:CA stoichiometry at saturation, we observed that the $K_D$ and off-rate for the CypA-capsid complex were only two-fold lower compared to complex with CA. Further, CypA binding resulted in only modest stabilization of the capsid (two-fold increase in half-life). Our data is consistent with a binding mode that relies primarily on an interaction via the canonical binding site and suggests that binding via the non-canonical binding site is either very weak and/or not immediately accessible for most CypA molecules interacting with the capsid. The fast interaction kinetics imply rapid turn-over of CypA and/or exchange with other host proteins that compete for binding to the capsid lattice.

The observation that CypA binding at low concentrations is neutral with respect to capsid opening provides an added dimension to our single-particle assay. The ability to 'paint' the capsid with fluorescent CypA at sub-stoichiometric ratios (about 1:20 CypA:CA) enables kinetic studies of capsid disassembly. We observed that defective capsids collapsed rapidly after membrane permeabilization suggesting that they are only stable in the presence of the high concentrations of free CA inside the viral membrane (most CA released by proteolysis during maturation is not incorporated into the capsid and remains in solution at a concentration in the low mM range). Similarly, we found that lattice

disassembly of an initially intact capsid is a rapid process that ensues as soon as the first gap in the lattice has appeared, that is uncoating is primarily controlled by the slow rate of removing the first (few) CA subunits.

The ability to simultaneously monitor appearance of the first defect and overall lattice stability allows us to define different stages of 'uncoating' and study how they may be influenced by external factors, as demonstrated here by dissecting the effects of two known capsid-binding molecules, PF74 and hexacarboxybenzene. PF74 is a drug molecule designed to target CA (*Blair et al., 2010*). At high concentration ($\geq$10 µM), it blocks HIV infection at a stage before reverse transcription (*Price et al., 2014*). It has been described as stabilizing self-assembled hexameric CA or CA-NC tubes but destabilizing HIV cores (*Bhattacharya et al., 2014*; *Da Silva Santos et al., 2016*; *Fricke et al., 2013*; *Shi et al., 2011*). In our assay, PF74 uncoupled the step of capsid opening from lattice disassembly. Addition of PF74 led to a pronounced acceleration of capsid opening but the remaining lattice was highly stabilized. This observation reconciles the previous ambiguity regarding the mode of action of PF74. It stabilizes CA lattices, but in a manner that makes it incompatible with a closed structure. Binding rapidly leads to capsid rupture in a manner that releases little CA but removes the barrier for free diffusion of proteins. In contrast, we found that hexacarboxybenzene is a potent inhibitor of capsid opening as seen from the remarkable increase in the half-life of this process. But in the event that the core does open, disassembly proceeds in a manner similar to the untreated capsid.

Taken together our observations suggest that the CA lattice has enhanced stability as long as it forms a closed surface, as predicted from coarse-grained simulations of capsid lattice disassembly (*Grime et al., 2016*). Appearance of a local defect leads to catastrophic failure of the lattice whereby the defect could rapidly propagate via dissociation of CA subunits that are no longer stabilized by dimer and/or trimer interfaces linking neighboring hexamers in the lattice (*Figure 7*). The interactions between subunits in a hexamer (or pentamer) represent a weak link in the CA lattice. Hexacarboxybenzene stabilizes the hexamer by binding in the its central electropositive pore (*Jacques et al., 2016*), leading to an increase in the energy barrier for dissociation of the first CA molecule(s), but only if the surrounding lattice is intact. Progressive loss of hexamer subunits would be expected to destroy the binding site for hexacarboxybenzene, leading to its dissociation and further destabilization of the disassembling part of the lattice. PF74 on the other hand binds in a pocket between CA subunits in a hexamer and stabilizes the hexameric lattice (*Bhattacharya et al., 2014*; *Price et al., 2014*). We speculate that PF74 may stabilize a particular conformation that is incompatible with the highly curved section of the capsid, leading to breakage at those points, whereby the restructured lattice is sufficiently stable to prevent dissociation of CA from hexamers adjacent to the gap.

What are the implications of our findings for uncoating in the cell? Two currently debated, non-mutually exclusive models for uncoating pose that core opening occurs either en route towards the nucleus (cytoplasmic uncoating) or after docking of the core at the nuclear pore complex (*Campbell and Hope, 2015*). In both models, residual CA protein remains associated with the pre-integration complex during and after nuclear import, but to what extent CA protein is lost from the capsid in the cytoplasmic uncoating model remains unclear. The intrinsic instability of the lattice after capsid opening observed here suggests that partially uncoated states with the capsid largely intact do not exist unless stabilized by host cell proteins. Such a stabilizing role has been suggested for CypA (*Liu et al., 2016*; *Shah et al., 2013*); however, the only modest stabilization of the lattice observed here in the presence of physiological concentrations suggests that other factors may be involved in this process. While not naturally present during an infection, the effect of the capsid-binding molecules, PF74 and hexacarboxybenze, offer insight into additional mechanisms that may be at play. PF74 binds at a junction between two neighboring CA proteins within the hexamer. This site can also be occupied by two host proteins: CPSF6 and Nup153 (*Matreyek et al., 2013*; *Price et al., 2014*). The observation that small-molecule binding at this site results in what could be described as a partially uncoated state raises the possibility that these two cofactors may also play roles in regulating the core-opening/uncoating event in vivo. In addition, the inhibition of capsid opening in the presence of hexacarboxybenze suggests that polyanionic species bound at the Arg18 cluster in the central pore of a hexamer could greatly enhance the lifetime of the closed core as it travel through the cytoplasm. Rapid uncoating could then be initiated, for example after docking at the nuclear pore complex, by catalyzing removal of a few CA subunits. In an accompanying paper,

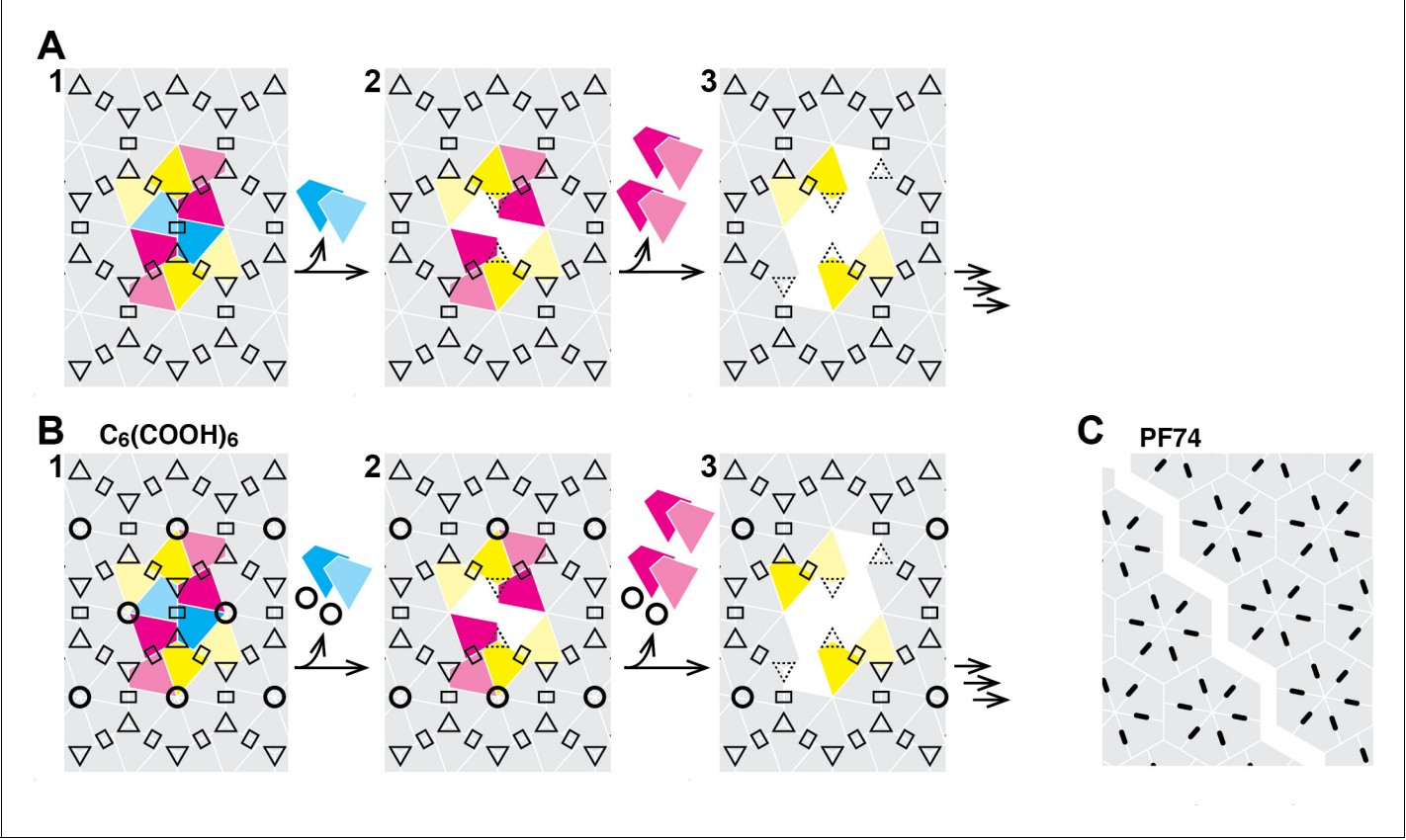

**Figure 7.** Schematic of the hexameric CA lattice and proposed disassembly pathway. (**A**) Intact CA lattice with dimeric and trimeric interfaces between hexamers highlighted as rectangles and triangles, respectively. Removal of a CA subunit (dark cyan) from a hexamer in the lattice leads to destabilization and dissociation of its dimer partner (light cyan) in the neighboring hexamer; GFP release may occur as early as at this stage. The defect propagates by dissociation of subunits (dark magenta) that have now lost contacts within the hexamer and at the trimer interface, followed by (or concomitant with) loss of their corresponding dimer partners (light magenta). Disassembly continues in all directions from the initial defect with increasing rate while the length of the defect edge grows. (**B**) Binding of hexacarboxybenzene (circles) to the central pore stabilizes the hexamer, reducing the probability of dissociation of the first CA subunit(s). Once the defect is created, the stabilizing effect of hexacarboxybenzene is insufficient to prevent disassembly via the pathway described in A. (**C**) Binding of PF74 (black lines) may lead to capsid rupture instead of loss of CA subunits, whereby PF74 molecules bound at the interfaces between CA subunit act as a glue to prevent dissociation of CA subunits from hexamers adjacent to the defect.

DOI: https://doi.org/10.7554/eLife.34772.017

we show that the naturally occurring polyanion inositol hexakisphosphate is selectively packaged into viral particles and stabilizes the mature lattice against spontaneous opening as observed here for the non-natural compound hexacarboxybenzene.

The competition for binding to the capsid between host molecules that facilitate or restrict infection, and their effect on uncoating remain to be fully determined, but our assay is ideally suited to answering these questions. The advantages of our approach include the ability to extract hundreds of single-particles traces in a single experiment for detailed kinetic analysis under defined conditions and the low requirements for reagents. The compatibility of the assay with lysates should prove powerful to dissect the effect of capsid-binding proteins and small molecules in complex mixtures containing components of different pathways and allows the study of perturbations via knock-down or overexpression of candidate proteins and lysates from different cell types or cells stimulated with different treatments, for example to induce antiviral states. The methods described here could readily be extended to competition binding studies or screening of the effect of small molecules on properties of the viral capsid for drug discovery.

# Materials and methods

**Key resources table**

| Reagent type (species) or resource | Designation | Source or reference | Identifiers | Additional information |
|---|---|---|---|---|
| Recombinant DNA reagent | pNL4.3-iGFP-ΔEnv | DOI: 10.1371/journal. ppat.1002762 | | |
| Recombinant DNA reagent | pNL4.3-iGFP-CA E45A-ΔEnv | This paper | | Proviral construct generated by subcloning a fragment with the CA E45A mutation into pNL4.3-iGFP-ΔEnv using the XbaI-SbfI restriction sites. |
| Recombinant DNA reagent | psPAX2 | NIH AIDS Reagent Program | #11348 | |
| Cell line (human) | HEK293T | ATCC | ATCC CRL-3216 | |
| Cell line (human) | HeLa | ATCC | ATCC CCL-2 | |
| Chemical compound, drug | PF74 | Sigma Aldrich | SML0835 | |
| Chemical compound, drug | BI2 | Enamine | Z1982491200 | |

## Constructs for production of viral particles

The proviral construct pNL4.3-iGFP-ΔEnv was generated as described previously (*Aggarwal et al., 2012*). It contains the open reading frame for eGFP flanked by protease cleavage sites inserted into the Gag gene between the coding sequences for MA and CA. In addition, it contains the SalI-BamHI fragment from HXB2, in which the start codon of the Env gene is mutated to a stop codon (TGA) to prevent expression of the envelope protein. The proviral construct pNL4.3-iGFP-CA E45A-ΔEnv containing an additional mutation to produce CA E45A was generated by subcloning a fragment with that mutation into pNL4.3-iGFP-ΔEnv using the XbaI-SbfI restriction sites. Constructs with PTAP to LIRL mutation in the late domain of Gag were generated by moving the SphI-SbfI fragment of a pNL4.3 construct with Gag-LIRL (courtesy of Eric Freed) into pNL4.3-iGFP-ΔEnv and into psPAX2. The second-generation lentiviral packaging plasmid psPAX2 (Cat# 11348, from Didier Trono) encoding the Gag, Pol, Rev, and Tat genes was obtained through the NIH AIDS Reagent Program, Division of AIDS, NIAID, NIH.

## Cell lines

HEK-293T cells and HeLa cells were obtained from ATCC. Identity testing was carried out by PCR. Cell lines tested negative for mycoplasma.

## Production, purification and biotinylation of viral particles

Viral particles were produced by transfecting HEK293T cells with a mixture of pNL4.3-iGFP-ΔEnv and psPAX2 (1.4:1, mol/mol) using polyethylenimine (PEI Max) reagent (Polysciences, 9002-98-6). The virus-containing medium was collected 48 hr post-transfection and centrifuged (2100 x g, 20 min, 4°C) to remove cells. Next, viral membrane proteins were biotinylated to allow immobilization of viral particles on surfaces. The biotinylation reagent EZ-Link Sulfo-NHS-LC-LC-Biotin (Thermo Scientific, 21338) was added to the virus-containing medium (~1 mg per 7 mL of medium) and the mixture was incubated for 90 min at 4°C. Biotinylated viral particles were then purified by size exclusion chromatography using a HiPrep 16/60 Sephacryl S-500 HR column (GE Healthcare) equilibrated with HBS (50 mM HEPES pH 7.5, 100 mM NaCl) and stored at 4°C for use within 7 days.

## Counting of GFP-positive viral particles

PTAP mutant virus was produced alongside WT pNL4.3-iGFP-ΔEnv as outlined in the methods above. To enumerate HIV particles per mL, 5 µL of each virus preparation was added to 45 µL of DMEM complete media and spinoculated onto poly-L-lysine-coated glass-bottom (175 µm thickness) 96-well plates (Greiner Sensoplate, Sigma) at 1200 × g for 60 min at 4°C. GFP-positive particles were detected using an inverted Olympus IX-70 microscope equipped with a 60 × oil immersion

objective (1.42 NA), microtiter stage (DeltaVision ELITE Image Restoration Microscope, Applied Precision/Olympus) and Evolve 512 EMCCD camera (Photometrics). Four fields of view were acquired per sample and counted using a viral particle count algorithm developed in Cellprofiler (Broad Institute) based on intensity and size cut offs. Viral counts per mL in the initial sample were then calculated from the viral count per $\mu m^2$ by taking the surface area of the well bottom and the initial sample volume into account. Means and standard deviations represent three independent viral counts.

## Expression and purification of CypA

Human CypA was expressed using a pET expression vector in BL21 (DE3) *E. coli* grown in Luria-Bertani medium at 37°C with shaking. Protein expression was induced by addition of isopropyl β-D-thiogalactopyranoside to a final concentration of 1 mM, and the cells were grown for 3 hr at 37°C with shaking. Cells were harvested by centrifugation, resuspended in cold (4°C) lysis buffer (25 mM HEPES, pH 7.6, 1 mM DTT, 0.02% NaN3, supplemented with 'Complete' protease inhibitor and 1 mg mL$^{-1}$ lysozyme) and lyzed by sonication on ice. The lysate was clarified by centrifugation (Thermo Fisher SS-34 rotor, 19,000 rpm, 30 min, 4°C). The supernatant was passed through a syringe filter (0.22 μm) and purified by anion exchange chromatography using a 10 mL HiTrap Q HP column (GE Healthcare Life Science, 17-1154-01) equilibrated with a buffer containing 25 mM HEPES, pH 7.6, 1 mM DTT, 0.02% NaN3. CypA eluted in the flow-through and fractions containing CypA were pooled and adjusted to a pH of 5.8 with 1% v/v acetic acid. After removal of aggregates by centrifugation (Sorvall SS34 rotor, 19,000 rpm, 1 hr, 4°C), the supernatant was applied to a cation exchange chromatography column (5 mL HiTrap SP HP, GE Healthcare Life Science, 17115201) equilibrated with a buffer containing 25 mM sodium phosphate, pH 5.8, 1 mM DTT, 0.02% NaN3 and CypA was eluted with a linear gradient from 0 to 1 M NaCl over 20 column volumes. Fractions containing purified CypA were dialyzed against CypA storage buffer (25 mM MOPS, pH 6.6, 1 mM DTT, 0.02% NaN3), concentrated using an Amicon-15 Ultra centrifugal filtration device (10 k MWCO, Merck, UFC901024) to a final concentration of 300 μM and frozen in liquid nitrogen for storage at −80°C. The yield was approximately 3 mg/g of cell mass.

## Labeling of CypA

CypA was labeled by reaction with a 4-fold molar excess of Alexa-Fluor 568-C5-maleimide (Thermo Fisher Scientific, A10254) in PBS (pH 7.4, 0.1 mM TCEP) for 10 min at room temperature. The reaction was quenched by addition of 1 M DTT to a final concentration of 17 mM. Labeled CypA was separated from unconjugated dye using Zeba desalting spin columns (Thermo Fisher Scientific, 89883) equilibrated with AF568-CypA storage buffer (50 mM Tris, pH 7.9, 20% v/v glycerol, 1 mM DTT). Under these conditions, CypA is quantitatively labeled at residue C51. CypA labeled at C51 binds in surface plasmon resonance measurements to immobilized CA or CA hexamers with the same affinity as unlabeled CypA. Labeled CypA was frozen in liquid nitrogen and stored at −40°C.

## Negative staining transmission electron microscopy of viral particles treated with perfringolysin O

Recombinant perfringolysin O was produced as described previously (*Rossjohn et al., 1997*). Viral particles purified by size exclusion chromatography were concentrated by 40-fold using Amicon Ultra-4 centrifugal ultrafiltration devices (Merck, UFC800324) and incubated with PFO at a final concentration of 0.5 μM at 37°C for 10 min in the presence of 0.1 mM inositol hexakisphosphate. A formvar/carbon grid (Ted Pella, 01801) was cleaned in a glow discharge unit and modified with Cell-Tak adhesive (0.05 mg mL$^{-1}$, Corning, 354240). The grid was then incubated with a sample (5 μL) containing PFO-treated viral particles for 1 min at room temperature, and the solution was wicked away using filter paper. The sample was immediately stained twice with uranyl formate (1% w/v) and excess solution wicked away with filter paper. The grids were dried under a gentle stream of nitrogen gas and allowed to dry further overnight. The samples were imaged using a FEI Tecnai G2 20 Transmission Electron Microscope equipped with a BM Eagle digital camera. Diameters of the PFO pores were measured using ImageJ.

## Fabrication and operation of microfluidic flow cells

Glass coverslips were cleaned by sonication in ethanol for 30 min followed by sonication in 1 M NaOH for 30 min, rinsed with ultrapure water and dried. PDMS devices for assembly of microfluidic flow cells (channel height 60 μm, channel width 800 μm) were prepared using standard protocols for soft lithography. After treating the PDMS device and the coverslip with an air plasma inside a plasma cleaner for 3 min, the PDMS device was mounted on the coverslip and the assembled microfluidic flow cell was heated in an oven at 70°C for at least 15 min to improve bonding between the glass and the PDMS. The glass surface at the bottom of the microfluidic channels was then modified by adsorption of a co-polymer composed of poly-L-lysine (PLL) and biotinylated poly(ethylene glycol) (PEG) (Susos AG, PLL(20)-g[3.4]-PEG(2)/PEG(3.4)-biotin (20%)). A solution of PLL-PEG-biotin (1 mg mL$^{-1}$ in PBS) was injected into the flow channels and incubated at room temperature for 30 min followed by flushing the channels with water and drying. The channels were then filled with a solution of streptavidin (Sigma-Aldrich, 0.2 mg mL$^{-1}$ in blocking buffer containing 20 mM Tris pH 7.5, 2 mM EDTA, 50 mM NaCl, 0.03% NaN3, 0.025% Tween 20, 0.2 mg mL$^{-1}$ BSA) for 15 min and rinsed with HBS pH 7.5. The microfluidic flow cell was mounted on the microscope stage and connected to tubing. Solutions were pulled through the channels at a flow rate of 100 μL min$^{-1}$ (unless specified otherwise) using a syringe pump connected to the outlet tubing and operating in 'withdraw' mode.

## TIRF microscopy assays

Biotinylated viral particles in HBS (30–100 μL) were flowed through the flow channel and captured on the modified coverslip surface. Unbound viral particles were washed out with 50 μL of HBS pH 7.5. TIRF microscopy assays were carried out with imaging buffer (50 mM HEPES pH 7.0, 100 mM NaCl), which was supplemented with an oxygen quenching system for dual colour experiments to reduce photobleaching (2 mM trolox, 2.5 mM protocatechuic acid, 0.25 U mL$^{-1}$ protocatechuate-3,4-dioxygenase).

Images were collected on a custom built TIRF microscope based around an ASI-RAMM frame (Applied Scientific Instrumentation) with a Nikon 100 × CFI Apochromat TIRF (1.49 NA) oil immersion objective. Lasers were incorporated using the NicoLase system (*Nicovich et al., 2017*). Images were captured on two Andor iXon 888 EMCCD cameras (Andor Technology Ltd). 300 mm tube lenses were used to give a field of view of 88.68 μm × 88.68 μm. Alternatively, images were collected on a TILL Photonics TIRF microscope equipped with a Zeiss 100 × Plan Apochromat (1.46 NA) oil immersion objective, solid state lasers for excitation, a beam splitter for simultaneous dual channel acquisition of fluorescence emission using two Andor iXon 897U EMCCD cameras for detection with a field of view of 46 μm × 46 μm.

### Capsid opening assay

Imaging buffer (30 μL) containing PFO (200 nM) was injected into the channel to initiate viral membrane permeabilization and TIRF images were acquired (488 nm laser, 20 ms exposure time) with a frequency of 1 frame s$^{-1}$ for the initial phase of the reaction (typically 300–500 frames) followed by a frequency of 0.1 frame s$^{-1}$. The total duration of the experiment varied depending on the uncoating kinetics, lasting between 15 and 90 min, whereby the total number of frames was kept at 800–900 frames.

### Capsid uncoating assay via painting with CypA

Imaging buffer (30 μL) containing PFO (200 nM) and Alexa Fluor 568-labeled CypA (1 μM) was injected into the flow channel. TIRF images were acquired sequentially for the different fluorophores (488 nm and 561 nm laser, 1–3 W cm$^{-2}$ power density, ~200 nm penetration depth, 20 ms exposure time) and a frequency of 1 frame s$^{-1}$. The following equation was used for model fitting to obtain estimates for the dissociation constant and the number of molecules bound at saturation (*Figure 3E*): N(eq) = [CypA]×N(max) / ([CypA]+K$_D$), where N(eq) is the number of molecules bound at equilibrium for a given CypA concentration, [CypA] is the concentration of CypA, N(max) is the number of molecules bound at saturation and K$_D$ is the dissociation constant.

Protein-capsid interaction assay

Protein binding was imaged (sequential acquisition, 488 nm and 561 nm laser, 20 ms exposure time, one frame s$^{-1}$) after flowing imaging buffer (30 μL) containing PFO (200 nM) and different concentrations of Alexa Fluor 568-labeled CypA (0.1–20 μM) into the flow channel. Protein dissociation was imaged (561 nm laser, 20 ms exposure time, five frame s$^{-1}$) after wash-out of labeled protein with imaging buffer.

## Image analysis

Images were analyzed with software written in MATLAB (The MathWorks, Inc.) adapted from previous work (*Böcking et al., 2011*). Images in the time series were aligned by cross-correlation to the first frame to correct for x/y-drift (*Guizar-Sicairos et al., 2008*). GFP-loaded viral particles appear as diffraction limited objects and were detected as local maxima in the first frame and their positions determined by Gaussian fitting. Overlapping particles were excluded from analysis. Fluorescence traces were calculated by integrating the fluorescence intensity in a 7 × 7 pixel region for each channel. The fluorescence intensity for each object was corrected for background fluorescence determined from pixels in the vicinity of the object. Fluorescence traces were fitted with step traces to identify the presence and time of steps corresponding to permeabilization and capsid opening. Traces were automatically sorted into four classes on the basis of the following criteria: (1) loss of entire GFP signal in one step; (2) loss of GFP intensity in one large (permeabilization) and one small (capsid opening) step; (3) loss of the majority of the GFP signal in one step with residual GFP signal persisting for the rest of the experiment; (4) no permeabilization or otherwise uninterpretable traces (excluded from analysis). The assignment to classes was verified by visual inspection of traces. Capsid opening times were calculated for traces in class two as the time difference between permeabilization and capsid opening. The time of reagent addition (or reagent wash-out) was detected as an overall increase (or decrease) of the background fluorescence in the reagent channel. The number of bound AF568-CypA molecules was determined from the ratio of the AF568-CypA fluorescence intensity associated with the capsid at equilibrium to the fluorescence intensity of a single AF568-CypA molecule. The fluorescence intensity of the single fluorophore was determined from the quantal photobleaching step in photobleaching traces of AF568-CypA molecules adsorbed sparsely to the coverslip surface and imaged continuously.

## Surface Plasmon resonance analysis of CypA binding to CA

### Biotinylation of CA

CA K158C was expressed and purified using methods adapted from the literature (*Hung et al., 2013*) and biotinylated at the engineered cysteine residue using the following procedure. CA K158C was first assembled into tubes in high salt buffer (50 mM Tris pH 8, 2.5 M NaCl, 0.1 mM TCEP, 0.02% sodium azide), collected by centrifugation (18,000 g, 5 min) and the pellet was resuspended in high-salt buffer. This centrifugation/resuspension procedure was repeated another two times. Biotin maleimide (Sigma-Aldrich B-1267) was then added in a twofold molar excess to the resuspended CA K158C tubes, allowed to react for one min and quenched by the addition 2-mercaptoethanol to a final concentration of 50 mM. The biotinylated tubes were washed by two cycles of centrifugation and resuspension in high-salt buffer as above. Finally, the tubes were collected by centrifugation, resuspended in buffer without salt to induce disassembly, and biotinylated CA was buffer exchanged into SPR running buffer (10 mM HEPES pH 8, 0.005% Tween 20, 2 mM EDTA, 100 mM NaCl) using a Zeba gel filtration spin column.

### Biotinylation of cross-linked CA hexamers

Soluble CA hexamers were assembled from CA A14C/E45C/W184A/M185A using a published protocol (*Pornillos et al., 2010*) and biotinylated using EZ-Link sulfo-NHS-LC-LC-biotin (Thermo Fisher Scientific, 21338). The biotinylation reagent was added at a twofold molar excess to CA hexamers in PBS (pH 7) and incubated at room temperature for 30 min. The reaction was stopped by buffer exchange into SPR running buffer using a Zeba gel filtration spin column.

## Surface chemistry and SPR measurements

Streptavidin was immobilized on the surface of a CM5 sensor chip (GE Healthcare, BR100399) using the amine coupling kit (GE Healthcare, BR100050). A solution containing EDC (7.5 mg mL$^{-1}$) and NHS (11.5 mg mL$^{-1}$) was injected into the flow cells to activate the carboxyl groups followed by a solution containing streptavidin (0.5 mg mL$^{-1}$) and finally ethanolamine (1 M) to block remaining activated carboxyl groups. All solutions were flowed through the channels for 7 min at a flow rate of 10 µL min$^{-1}$. Finally, biotinylated CA K158Cor biotinylated CA hexamer were immobilized on the surface by injection a 2.5 µM solution for 30 s at a flow rate of 30 µL min$^{-1}$. Flow channels modified with streptavidin were used as reference cells. CypA was buffer exchanged into SPR running buffer using a Zeba gel filtration spin column. CypA solutions at a range of concentrations (3.125–100 µM) were flowed through the cells (20 s at a flow rate of 100 µL min$^{-1}$) followed by a buffer wash (30 s at a flow rate of 100 µL min$^{-1}$) while measuring the SPR response at a frequency of 40 Hz.

## Preparation of cell lysate

HeLa cells were washed with PBS, resuspended in lysis buffer (10 mM Tris-HCl, pH 8, 10 mM KCl, 1.5 mM MgCl2, protease inhibitor and phosphatase inhibitor), and incubated for 15 min at 4°C. Lysis was achieved by passing the cell suspension through a 30.5 g needle (130 strokes with cooling the sample on ice for 2 min at every 10th stroke). Cell lysis was verified by staining a sample of the lysate with trypan blue (added to a final concentration 0.2% w/v). The lysate was centrifuged (18,000 x g, 30 min, 4°C) to remove cell debris and protein levels in the supernatant were estimated from the absorbance at 280 nm. Lysate was frozen in liquid nitrogen and stored at −40°C.

## Acknowledgements

We thank Nick Dixon (University of Wollongong) for supplying the construct for expression of CypA and Nick Ariotti (UNSW) for technical assistance and use of facilities at the Electron Microscope Unit at UNSW. We thank Sara Lawrence (St. Vincent's Institute) for purification of recombinant PFO, Jesse Goyette (UNSW) for help with SPR measurements, Amir Mousapasandi (UNSW) for help with labeling CypA and Alex McLeod (UNSW) for help with data analysis. We thank Johnson Mak (Griffith University) for discussions and critical feedback. CM and DL received an Australian Government Research Training Program Scholarship. TB received funding from the Australian Centre for HIV and Hepatitis Virology Research. TB and ST received funding from the National Health and Medical Research Council of Australia (NHMRC APP1100771). Funding from the Victorian Government Operational Infrastructure Support Scheme to St Vincent's Institute is acknowledged. MWP is a National Health and Medical Research Council of Australia Research Fellow.

## Additional information

### Funding

| Funder | Grant reference number | Author |
| --- | --- | --- |
| National Health and Medical Research Council | APP110071 | Stuart Turville<br>Till Böcking |
| Australian Centre for HIV and Hepatitis Virology Research | | Till Böcking |

The funders had no role in study design, data collection and interpretation, or the decision to submit the work for publication.

### Author contributions

Chantal L Márquez, Formal analysis, Investigation, Writing—review and editing; Derrick Lau, Vaibhav Shah, Conall McGuinness, Investigation, Writing—review and editing; James Walsh, Software, Investigation, Writing—review and editing; Andrew Wong, Anupriya Aggarwal, Investigation; Michael W Parker, Resources, Writing—review and editing; David A Jacques, Writing—original draft, Writing—review and editing; Stuart Turville, Conceptualization, Resources, Writing—review and editing; Till Böcking, Conceptualization, Formal analysis, Writing—original draft, Writing—review and editing

## Author ORCIDs
Michael W Parker (ID) http://orcid.org/0000-0002-3101-1138
David A Jacques (ID) http://orcid.org/0000-0002-6426-4510
Till Böcking (ID) http://orcid.org/0000-0003-1165-3122

## Decision letter and Author response
Decision letter https://doi.org/10.7554/eLife.34772.020
Author response https://doi.org/10.7554/eLife.34772.021

## Additional files

### Supplementary files
• Transparent reporting form
DOI: https://doi.org/10.7554/eLife.34772.018

### Data availability
All data generated or analysed during this study are included in the manuscript and supporting files.

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
