## [Decision Letter]

Thank you for submitting your article "Kinetics of HIV-1 capsid uncoating revealed by single-molecule analysis" for consideration by *eLife*. Your article has been reviewed by two peer reviewers, and the evaluation has been overseen by Wes Sundquist as the Reviewing Editor and Wenhui Li as the Senior Editor. The following individual involved in review of your submission has agreed to reveal their identity: Hans-Georg Kräusslich.

The reviewers have discussed the reviews with one another and the Reviewing Editor has drafted this decision to help you prepare a revised submission.

Márquez et al. describe a novel single-molecule imaging approach for visualizing HIV-1 capsid uncoating. Capsid uncoating after cell entry is still enigmatic and hotly debated in the field, and no clear picture exists, so the manuscript represents a timely contribution. The authors demonstrate that virus particles engineered to encapsidate the fluorescent marker GFP can be immobilized on a coverslip and then efficiently permeabilized with a bacterial pore-forming protein. Importantly, the authors demonstrate that the permeabilization process does not completely disrupt the lipid envelope and that the viral cores are retained inside the partially disrupted membrane. The authors are then able to use loss of GFP signals to discern 3 subpopulations of viral cores – leaky, opening, or closed. To demonstrate the utility of their assay, the authors analyze the core populations over time in the presence or absence of a variety of co-factors or CA mutations previously published to have either positive or negative effects on core stability. These initial experiments quantified the kinetics of GFP release (i.e., "capsid opening") over time. Using this readout, the authors recapitulate the expected effects of the different cofactors and the stabilizing E45A mutation on core stability.

In the second half of the manuscript, the authors describe another technical advance in which they use fluorescent-labelled CypA binding (CypA "paint") to characterize the kinetics of lattice disassembly (i.e., the fate of the bulk lattice following core opening). The authors use this approach to discern the mechanisms of action of PF74 and hexacarboxybenzene on virion cores. Interestingly, the authors find that while PF74 increases the rate of core opening, the capsid lattice remains intact. Hexacarboxybenzene, on the other hand, slows core opening considerably, but lattice integrity is rapidly compromised once core opening does occur. Finally, the authors demonstrate that their assay is compatible with analyses of the effects of cell lysate, which they find slows capsid opening.

This study represents a significant technical advance in the biochemical analysis of HIV-1 uncoating, the study makes nice use of several relevant small molecules, and the imaging and quantification of large numbers of individual events (rather than bulk studies or following just a few particles inside cells) adds significantly to our understanding and may help to clarify outstanding issues. The experiments are generally well executed, but several aspects need to be clarified or controlled.

Significant Issues that need to be addressed:

1) The text refers to immobilized viral particles, but the preparation, as described, will not clearly separate virus-like particles from membrane vesicles (and both of these species will be biotinylated and captured). Accordingly, the immobilized objects may be a mixture, although the virus-like particles should, at least in principle, have greater GFP fluorescence (incorporated via Gag). However, the permeabilized particles, as seen in the EM images, seem rather large for an HIV particle preparation (it's difficult to measure on the image, but some appear to be >200nm). It is therefore not clear what fraction of these objects are HIV and what fraction are vesicles (and if vesicles are present and being scored, then these would presumably lead an over estimate of the fraction of leaky capsids). The basic experiment should therefore be repeated with viron preparations that have been treated with subtilisin (to remove any vesicles). It is also a bit puzzling why conical capsids are not visible by negative staining inside the permeabilized virus-like particles. The authors should also report the GFP intensity variation in the initial fields of particles.

2) The major part of the manuscript serves to describe and validate the assay. This sometimes reads a bit like a circular argument. As there is no direct proof that the signals remaining are indeed intact or partially disassembled capsids (in the absence of direct visualization by e.g. EM), the authors correctly use other tools to validate what they are (i.e. stabilizing molecules or stabilizing mutations etc). This is appropriate, but should then not be used to say that observations confirm these effects. If a known feature is used to validate the assay, observing the expected phenotype does not confirm the feature. Accordingly, the majority of the figures serve to validate the assay and its implications for capsid opening and disassembly. This is fine, but should be stated more clearly.

3) The authors argue for the biological relevance of the subset of capsids that uncoat with some delay are relevant, whereas those that fail to uncoat over the time period would be too stable. However, we really don´t understand the role of the host cell environment, so this seems an overstatement. The experiment with diluted HeLa cytoplasmic extract is a nice initial result, but this is not a relevant cell type and the appropriate host factors may not be present at the correct levels (to say nothing of potential changes with cell type, cellular substructures such as microtubule motor forces, etc.).

Other issues for the author's consideration:

1) Subsection “Single-particle fluorescence imaging of capsid opening”, first paragraph, the statement that PFO-mediated permeabilization mimics some aspects of viral fusion seems a bit reaching. While the system described in the manuscript is very nice, introducing (defined) holes into a membrane is rather removed from the process of viral fusion. It doesn't add anything significant to the paper, so the authors may consider removing the statement.

2) Can the k1 and k2 curves, in addition to the Figure 2B sum plots, be reported (or at least report the fractions of particles in the "fast" and "slow" decay time populations)?

3)Please justify with a more detailed calculation the statement that "This fraction was consistent with the amount of GFP expected to be randomly trapped inside the volume of the capsid (relative to the volume of the viral particle) during maturation."

4) There is no callout to Figure 3F. Should this, and callouts to Figure 3 figure supplements, belong in the last paragraph of the subsection “Kinetics of protein binding to intact HIV capsids”?

5) Discussion, seventh paragraph. Please provide citation(s) after Nup153.

6) Subsection “Labeling of CypA”. Please indicate whether you verified that CypA, labelled at C51, binds CA with the same affinity as unlabeled CypA.

7) The bibliography needs to be cleaned up – at least 5 citations lack volume/ page indicators. In addition, several of the references in the Introduction should be updated with more recent references.

8) Throughout the manuscript, virus-like particles carrying GFP within the Gag polyproteins are used. While this may not influence mature capsid formation or stability, this possibility should at least be acknowledged.

9) The authors refer in various parts to binding to the CA monomer vs. hexamer, but CA at the indicated concentrations would be a dimer (unless constructed to form the hexamer).

10) It makes sense to use PF74 at the relatively high concentration, where it is supposed to affect capsid integrity, but it would also be informative if this experiment were performed at the low concentration, where others have argued that the compound only affects cofactor binding without disturbing capsid architecture.

11) Subsection “Kinetics of protein binding to intact HIV capsidsKinetics of protein binding to intact HIV capsids”, last sentence, should read "CA" rather than "CypA".

Additional statistical issues:

1) Although some datasets harbor error bars, the paper does a rather poor job describing numbers of biological replicates (independent experiments) and technical replicates (number of replicates within individual experiments). Although single-molecule datasets comprise many individual measurements, biological replicates surely still apply. The easiest way to supply this transparency would be to state in the figure legends how many replicates were done.

2) Figure 2A lacks error bars, and thus may be representative of one of x number of replicates (as mentioned, please supply such info). Also, please define error bars (SEM, SD, etc).

3) The paper lacks statistical considerations of results differences. For example, is the influence of 20 μM CypA in Figure 2C significant? It would seem that something like a t test analysis will readily address this question. Authors should more comprehensively analyze results statistically.

---

## [Author Response]

Significant Issues that need to be addressed:1) The text refers to immobilized viral particles, but the preparation, as described, will not clearly separate virus-like particles from membrane vesicles (and both of these species will be biotinylated and captured). Accordingly, the immobilized objects may be a mixture, although the virus-like particles should, at least in principle, have greater GFP fluorescence (incorporated via Gag). However, the permeabilized particles, as seen in the EM images, seem rather large for an HIV particle preparation (it's difficult to measure on the image, but some appear to be >200nm). It is therefore not clear what fraction of these objects are HIV and what fraction are vesicles (and if vesicles are present and being scored, then these would presumably lead an over estimate of the fraction of leaky capsids). The basic experiment should therefore be repeated with viron preparations that have been treated with subtilisin (to remove any vesicles). It is also a bit puzzling why conical capsids are not visible by negative staining inside the permeabilized virus-like particles. The authors should also report the GFP intensity variation in the initial fields of particles.

We have addressed the reviewers' concerns about possible contributions from contaminating microvesicles in the viral preparation to the read-out of the capsid opening assay as follows:

Are vesicles being scored in the assay?

As a stringent control to rule out GFP-positive microvesicles, we have generated a viral mutant that is not released from producer cells through the mutation of the PTAP motif in the late domain of Gag (the motif that recruits the ESCRT pathway for particle abscission). If GFP-positive microvesicles were to form in our preparations, then these should be released from cells independent of virus particles. However, when removing the PTAP motif from the virus and helper constructs, we observe a 99.3% reduction in GFP-positive particle counts. This observation now gives us the confidence that GFP-positive signals are representative of virions and not related to other structures such as microvesicles.

Unfortunately, we could not treat virus preparations with subtilisin because it is incompatible with our assay. Subtilisin removes the membrane proteins required for biotinylation and capture of viral particles onto the surface of the microscope cover slip.

We have added the new data to the manuscript as Figure 1B.

“As a control we also used constructs containing a mutation in the late domain of Gag that leads to a block in the abscission of viral particles from the producer cell. The number of GFP-positive particles in the cell supernatant decreased by 99.7% when viral release was blocked (Figure 1B), confirming that essentially all GFP-positive particles released in the absence of the block represent viral particles.”

Why are conical capsids are not visible by negative staining inside the permeabilized virus-like particles?

We have acquired negative staining electron micrographs of viral particles treated with PFO in the presence of IP6, a polyanion that strongly stabilizes intact capsids (as shown in our accompanying paper). Under these conditions, we detect a subset of particles with ring-shaped pores that also contain conical capsids, demonstrating that PFO forms pores in the viral membrane. The images also contain empty membranes with ring-shaped pores, which presumably represent a mixture of (1) PFO-permeabilized viral particles after capsid disassembly (“leaky” capsids are not stabilised by polyanions and would be expected to disassemble rapidly upon membrane permeabilisation) and (2) PFO-permeabilized extracellular vesicles that are also contained in the preparation of viral particles. The new EM data has been added to the manuscript as Figure 1D.

The authors should also report the GFP intensity variation in the initial fields of particles.

Added to the manuscript as part of Figure 1G.

2) The major part of the manuscript serves to describe and validate the assay. This sometimes reads a bit like a circular argument. As there is no direct proof that the signals remaining are indeed intact or partially disassembled capsids (in the absence of direct visualization by e.g. EM), the authors correctly use other tools to validate what they are (i.e. stabilizing molecules or stabilizing mutations etc). This is appropriate, but should then not be used to say that observations confirm these effects. If a known feature is used to validate the assay, observing the expected phenotype does not confirm the feature. Accordingly, the majority of the figures serve to validate the assay and its implications for capsid opening and disassembly. This is fine, but should be stated more clearly.

We have restructured the section describing the effect of perturbations on capsid opening to better identify known features that are used for assay validation. We note however that most of our measurements offer additional insight beyond assay validation in that they distinguish between opposite effects reported in the literature and/or quantify the degree of capsid stabilisation/destabilisation. For example, the observation that CypA slows capsid disassembly is expected on the basis of the literature, but that it has only a modest effect on capsid half-life at cellular concentrations was not known before. Similarly, the observation that PF74 changes GFP release kinetics validates that we are measuring a feature of capsid uncoating, but distinguishing effects between capsid failure and lattice stabilization is a new discovery. Specifically, our single molecule data have reconciled two opposing observations in the field; namely that PF74 appears to stabilize hexamer tubes in vitro but cause phenotypes indicative of early uncoating in cells. Our data suggest that PF74 does stabilise but by rigidifying such that an opening event is induced. Fascinatingly, the remaining lattice is then stabilised against further dissociation.

3) The authors argue for the biological relevance of the subset of capsids that uncoat with some delay are relevant, whereas those that fail to uncoat over the time period would be too stable. However, we really don´t understand the role of the host cell environment, so this seems an overstatement. The experiment with diluted HeLa cytoplasmic extract is a nice initial result, but this is not a relevant cell type and the appropriate host factors may not be present at the correct levels (to say nothing of potential changes with cell type, cellular substructures such as microtubule motor forces, etc.).

We agree that it is premature to include or exclude the biological relevance of different stability classes and we have deleted the paragraph where we had speculated on the “correct” capsid stability required for infection.

Other issues for the author's consideration:1) Subsection “Single-particle fluorescence imaging of capsid opening”, first paragraph, the statement that PFO-mediated permeabilization mimics some aspects of viral fusion seems a bit reaching. While the system described in the manuscript is very nice, introducing (defined) holes into a membrane is rather removed from the process of viral fusion. It doesn't add anything significant to the paper, so the authors may consider removing the statement.

The statement has been removed.

2) Can the k1 and k2 curves, in addition to the Figure 2B sum plots, be reported (or at least report the fractions of particles in the "fast" and "slow" decay time populations)?

The number of particles in the "fast" and "slow" decay time populations (Figure 2B) are reported in the table in Figure 2—figure supplement 2.

3)Please justify with a more detailed calculation the statement that "This fraction was consistent with the amount of GFP expected to be randomly trapped inside the volume of the capsid (relative to the volume of the viral particle) during maturation."

We have added a detailed calculation for the volume of the capsid relative to the volume of the viral particle to the legend of Figure 1—figure supplement 1.

“The mean fraction of encapsidated GFP is ~13%, similar to the ratio of the volume inside the capsid to the total volume of the viral particle (16%), which was estimated on the basis of the following considerations:

Internal volume of the viral particle: The internal virus volume (i.e. excluding the viral membrane and matrix protein layer) has been estimated by cryo-electron tomography (cET) of HIV particles, giving an average value of particle=6.79×105nm3 [from Figure 4, (de Marco et al., 2012)]; this volume corresponds to an internal diameter of 109 nm. […] The fraction of the internal volume of the capsid relative to the total internal volume of the viral particle is given by Vcore internal/V particle=16%.”

4) There is no callout to Figure 3F. Should this, and callouts to Figure 3 figure supplements, belong in the last paragraph of the subsection “Kinetics of protein binding to intact HIV capsids”?

Callouts to Figure 3F (subsection “Visualizing capsid lattice disassembly using CypA paint”, first paragraph) and to Figure 3 figure supplements (subsection “Kinetics of protein binding to intact HIV capsids”) have been added to the manuscript.

5) Discussion, seventh paragraph. Please provide citation(s) after Nup153.

We have provided citations after Nup153.

6) Subsection “Labeling of CypA”. Please indicate whether you verified that CypA, labelled at C51, binds CA with the same affinity as unlabeled CypA.

We have carried out SPR experiments to verify that CypA, labelled at C51, binds CA with the same affinity as unlabeled CypA. This information has been added to the Materials and methods section.

“CypA labeled at C51 binds in surface plasmon resonance measurements to immobilized CA or CA hexamers with the same affinity as unlabeled CypA.”

7) The bibliography needs to be cleaned up – at least 5 citations lack volume/ page indicators. In addition, several of the references in the Introduction should be updated with more recent references.

We have added missing volume/page indicators and more recent references to the Introduction.

8) Throughout the manuscript, virus-like particles carrying GFP within the Gag polyproteins are used. While this may not influence mature capsid formation or stability, this possibility should at least be acknowledged.

We have added a section to the Discussion to point out that the viral particles used in this work carry GFP within the Gag polyproteins, and that this modification may affect mature capsid formation or stability.

“Here we have quantified the fraction of particles of different stability for a viral particle that lacks HIV envelope protein and has GFP inserted into the Gag polyprotein. […] It is therefore important to note that wild type HIV may exhibit a different distribution of capsid stability types.”

9) The authors refer in various parts to binding to the CA monomer vs. hexamer, but CA at the indicated concentrations would be a dimer (unless constructed to form the hexamer).

We have changed the manuscript to refer to the recombinant CA K158C immobilised on the SPR chip as “CA”. The immobilisation of CA on the SPR chip was carried out at a concentration of 2.6 μM (about 7-fold lower than the K_D_ for CA dimerization of ~18 μM). The immobilised species are likely to be a mixture of primarily monomers and some dimers.

10) It makes sense to use PF74 at the relatively high concentration, where it is supposed to affect capsid integrity, but it would also be informative if this experiment were performed at the low concentration, where others have argued that the compound only affects cofactor binding without disturbing capsid architecture.

We agree that dissecting out the relative effects on capsid of PF74 at different concentrations and the impact of this on direct uncoating, cofactor binding and cofactor induced changes could be of significant interest. We feel that this would be worthy of a comprehensive study of its own that is focused on this question and plan to focus on this in future work.

11) Subsection “Kinetics of protein binding to intact HIV capsidsKinetics of protein binding to intact HIV capsids”, last sentence, should read "CA" rather than "CypA".

We have corrected this mistake.

Additional statistical issues:1) Although some datasets harbor error bars, the paper does a rather poor job describing numbers of biological replicates (independent experiments) and technical replicates (number of replicates within individual experiments). Although single-molecule datasets comprise many individual measurements, biological replicates surely still apply. The easiest way to supply this transparency would be to state in the figure legends how many replicates were done.

The number of replicates for data sets have been added to the respective figure legends.

2) Figure 2A lacks error bars, and thus may be representative of one of x number of replicates (as mentioned, please supply such info). Also, please define error bars (SEM, SD, etc).

We have added the number of replicates for Figure 2A and defined error bars in the figure legends for Figure 2 and Figure 6.

3) The paper lacks statistical considerations of results differences. For example, is the influence of 20 μM CypA in Figure 2C significant? It would seem that something like a t test analysis will readily address this question. Authors should more comprehensively analyze results statistically.

We have added statistical analysis of the differences in Figure 2 and Figure 6.